# Adversarial Attacks on Cooperative Multi-Agent Bandits

## Abstract

Cooperative multi-agent multi-armed bandits (CMA2B) consider the collaborative efforts of multiple agents in a shared multi-armed bandit game. We study latent vulnerabilities exposed by this collaboration and consider adversarial attacks on a few agents with the goal of influencing the decisions of the rest. More specifically, we study adversarial attacks on CMA2B in both homogeneous settings, where agents operate with the same arm set, and heterogeneous settings, where agents may have distinct arm sets. In the homogeneous setting, we propose attack strategies that, by targeting just one agent, convince all agents to select a particular target arm $T - o(T)$ times while incurring $o(T)$ attack costs in $T$ rounds. In the heterogeneous setting, we prove that a target arm attack requires linear attack costs and propose attack strategies that can force a maximum number of agents to suffer linear regrets while incurring sublinear costs and only manipulating the observations of a few target agents. Numerical experiments validate the effectiveness of our proposed attack strategies.

## 1 Introduction

Cooperative multi-agent multi-armed bandits (CMA2B) have been widely studied in recent years (Bistritz & Leshem, 2018; Boursier & Perchet, 2019; Yang et al., 2021; Wang et al., 2023a). In CMA2B, $M \in \mathbb{N}^+$ agents cooperatively play multi-armed bandits with $K \in \mathbb{N}^+$ arms in a sequential manner. In each decision round, each agent picks one arm and observes a reward sample drawn from a stochastic distribution associated with the pulled arm. Their cooperative objective is to maximize their total cumulative rewards in $T \in \mathbb{N}^+$ decision rounds, or minimize their total regret—the difference between the total expected rewards of all agents constantly pulling the optimal arm and the actual expected rewards of the considered algorithm.

Leveraging cooperation between agents, CMA2B algorithms can achieve an improved total regret of $O(K \log T)$ (Wang et al., 2023a), compared with a total regret of $O(MK \log T)$ if no cooperation is used. However, a security caveat arises when some agents occasionally get unreliable observations that may have been tampered with by malicious attackers. This concern becomes more serious in large-scale multi-agent systems, where assuring consistent reliability of every agent's observations becomes increasingly intricate. Given the collaborative nature of CMA2B, such adversarial attacks have the potential to not only influence the performance of their target agents, but also affect other agents in the same learning system. For example, botnets can mimic user clicks on ads to mislead learning algorithms in online advertising. When advertisers, or learning agents, exchange their information such as the user's clicks for cooperative user preference learning, it can be seen as a CMA2B problem. The risks come from the malicious botnet, sitting between the users and the agents. The objective of the bot is to undermine algorithm's performance, however, due to the limit of resources, the bot hope to achieve this objective with as few operations, such as removing the user's clicks or adding fake ones, as possible.

Adversarial attacks on single-agent bandits have been studied in (Jun et al., 2018; Liu & Shroff, 2019; Zuo, 2024). In single-agent bandits, a successful attack means convincing the agent to pull a target arm a nearly linear number of times (i.e., $T - o(T)$) via manipulation of the agent's reward observations while only incurring a sublinear attack cost (i.e., $o(T)$). In contrast, in the multi-agent settings, the definition of a successful attack may vary. It might involve misleading a single agent or all agents, and the manipulations could target one agent or span across all agents. In this paper, we

Table 1: Summary of settings and attack strategies[†]

| Setting | Target Agents | Affected Agents | Attack Objective | Cost |
|---|---|---|---|---|
| Single-agent Jun et al. (2018) | 1 | 1 | target arm: $T - o(T)$ | $O(K\sqrt{\log T})$ |
| Homogeneous (Warm-up Result) | 1 | $M$ | target arm: $MT - o(T)$ | $O(K\sqrt{\log T})$ |
| Heterogeneous (Main Result) | $|\mathcal{K}_0|$ | $\sum_{k \in \mathcal{K}_0} M_*(k)$ | regret: $\sum_{k \in \mathcal{K}_0} M_*(k)T - o(T)$ | $O(|\mathcal{K}_0|\sqrt{\log T})$ |

[†] $\mathcal{K}_0$: output arms of AAS; $M_*(k)$: # of agents with local optimal arm $k$.

consider a challenging attack objective: *affecting the maximum number of agents via attacking the reward observations of only a small subset of agents*.

Pursuing this objective, we discuss adversarial attacks in both homogeneous and heterogeneous CMA2B contexts. In the homogeneous settings, where each agent has access to all $K$ arms, it is feasible to convince all agents to choose a specific target arm in most time slots (if the target is a suboptimal arm, agents suffer linear regret). However, the number of agents that must be attacked and the overall cost of these attacks remain uncertain (addressed in Section 3). Conversely, in the heterogeneous settings, where agents have different subsets of arms, the goal of directing all agents to select a single target arm becomes unattainable, especially if some agents lack access to this arm. As we elaborate in Section 4, even the task of convincing a subset of agents to select a target arm could require linear attack costs. Consequently, we shift our attack objective towards inducing the greatest number of agents to suffer linear regret. This objective leads to three new challenges. First, determining the largest group of affected agents experiencing linear regret while incurring only sublinear costs remains an unresolved issue. Second, it is unclear how to select a small number of target agents for attack. Third, for these chosen target agents, we need to design attack strategies that can effectively influence other agents.

**Our Contributions**. In this paper, we provide an in-depth study of adversarial attacks on CMA2B. In the homogeneous setting, we propose attack strategies that can convince all agents to select a designated target arm linear times by attacking only a single agent with sublinear attack costs, revealing the inherent vulnerability of homogeneous CMA2B algorithms. In the heterogeneous settings, we demonstrate that the target arm attack may demand linear attack costs, and propose attack strategies that can compel a significant number of agents to experience linear regret by manipulating the observations of only a few agents with sublinear attack costs. Table 1 summarizes attack strategies in different settings. Our technical contributions are outlined below.

- In Section 3, for homogeneous CMA2B, we devise attack strategies for three representative algorithms, CO-UCB (Yang et al., 2022), UCB-TCOM (Wang et al., 2023a), and DPE2 (Wang et al., 2020a), by targeting a *single* agent to misguide *all* agents. We prove their attack costs are independent of the number of agents $M$.

- In Section 4.1, for heterogeneous CMA2B, after illustrating examples in which linear costs are necessary to fool all agents into suffering linear regrets, we provide a criterion to determine whether two agents can be simultaneously misled with sublinear attack costs.

- In Section 4.2, we extend the above criterion to the Affected Agents Selection (AAS) algorithm that identifies the largest set of agents eligible to be affected with an approximation guarantee. Further, we design the Target Agents Selection (TAS) algorithm to choose small subsets of agents (target agents) to attack that can influence all agents selected by AAS.

- Based on AAS and TAS, we propose the Oracle Attack (OA) strategy tailored for heterogeneous environments with known arm rankings. Agent heterogeneity presents a significant attackability challenge, while we provide a non-trivial analysis showing our meticulous agent selection can effectively address it. In Section 4.3, we extend this strategy to unknown environments via the Learning-Then-Attack (LTA) strategy with its cost analysis.

In addition to the algorithmic and theoretical contributions, we conduct experiments to evaluate our proposed attack strategies. Due to the space limit, full proofs are included in the Appendix.

**Related Work**. There is a large literature of work focused on CMA2B, e.g., (Rosenski et al., 2016; Bistritz & Leshem, 2018; Boursier & Perchet, 2019; Wang et al., 2020a;b; Shi et al., 2021; Yang et al., 2021; 2022; Wang et al., 2022; 2023a;b) and the references therein. However, only a few works have studied misinformation in CMA2B learning, i.e., (Boursier & Perchet, 2020; Vial et al.,

2021; Madhushani et al., 2021; Dubey & Pentland, 2020). In these scenarios, there are either malicious/selfish agents (e.g., byzantine agents (Dubey & Pentland, 2020)) sharing wrong information (e.g., false arm recommendation (Vial et al., 2021), wrong reward observations (Boursier & Perchet, 2020)), or imperfect communication (e.g., adversarial corruption (Madhushani et al., 2021)), resulting in other agents failing to find the optimal arm. Our work is the first to study how an attacker may manipulate multi-agent cooperative learning.

Regarding adversarial attacks on single-agent bandit problems, a growing literature focuses on a variety of settings, e.g., (Jun et al., 2018; Garcelon et al., 2020; Ma & Zhou, 2023; Zuo et al., 2024; Liu & Lai, 2020; Wang et al., 2023c). Specifically, Jun et al. (2018) pioneered the formulation of adversarial attack models for stochastic bandits. Garcelon et al. (2020) delved into attacks on linear contextual bandits, while Ma & Zhou (2023) examined attacks on adversarial bandits. Notably, these investigations do not extend to multi-agent bandit scenarios. While Vial et al. (2021) explored a related concept with honest and malicious agents in collaborative bandit settings, to the best of our knowledge, we are the first to investigate the vulnerabilities of CMA2B under adversarial attacks.

## 2 PRELIMINARIES

We consider a CMA2B consisting of $K \in \mathbb{N}^+$ arms, denoted by an arm set $\mathcal{K} := \{1, 2, \cdots, K\}$, and $M \in \mathbb{N}^+$ agents, denoted by an agent set $\mathcal{M} := \{1, 2, \cdots, M\}$. Each arm $k \in \mathcal{K}$ is associated with a $\sigma^2$-sub-Gaussian reward distribution and unknown mean $\mu(k)$. We assume $\mu(1) \geqslant \cdots \geqslant \mu(K)$. There are $T \in \mathbb{N}^+$ decision rounds for CMA2B, denoted the round set by $\mathcal{T} := \{1, 2, \ldots, T\}$. We consider both homogeneous and heterogeneous settings for cooperative multi-agent bandits.

**Homogeneous settings**. In the homogeneous setting, each agent can equally observe and select every arm in $\mathcal{K}$. In round $t \in \mathcal{T}$, each agent selects an arm $k_t^{(m)} \in \mathcal{K}$ and observes a reward $X_t^{(m,0)}(k_t^{(m)})$ with expectation $\mu(k_t^{(m)})$, where the superscript $^{(m,0)}$ refers to the vanilla (pre-attack) reward observation on agent $m$. These agents share their information about arms with each other. We use regret to measure the performance of a policy, defined as

$$R(T) := MT\mu(1) - \sum_{m=1}^{M} \sum_{t=1}^{T} \mu(k_t^{(m)}),$$

which is the difference between the maximized accumulative reward (all agents keep pulling optimal arm 1) and the concerned policy's total reward. Note that we do not consider the collision (e.g., Boursier & Perchet (2019)) here, which means different agents can select the same arm in the same round, and each of them gets an independent reward sample. All agents together aim to minimize the regret.

**Heterogeneous settings**. In heterogeneous settings, each agent $m \in \mathcal{M}$ has access only to a subset of arms $\mathcal{K}^{(m)} \subset \mathcal{K}$. Assume $|\mathcal{K}^{(m)}| > 1$ for each agent $m \in \mathcal{M}$ to exclude trivial cases. For each agent $m$, we denote the arm with the highest reward mean in $\mathcal{K}^{(m)}$ as $k_*^{(m)} = \arg\max_{k \in \mathcal{K}^{(m)}} \mu(k)$, called agent $m$'s *local optimal arm*. Similarly to the homogeneous setting, after agent $m$ selects arm $k_t^{(m)}$ in round $t$, the environment also reveals a sub-Gaussian reward $X_t^{(m,0)}(k_t^{(m)})$ with expectation $\mu(k_t^{(m)})$. Agents share their information with others. However, due to different local optimal arms $k_*^{(m)}$ benchmarks, the regret is defined differently:

$$R(T) = T \sum_{m=1}^{M} \mu(k_*^{(m)}) - \sum_{m=1}^{M} \sum_{t=1}^{T} \mu(k_t^{(m)}).$$

**Threat model**. In both the homogeneous and heterogeneous scenarios, agent $m$ selects an arm $k_t^{(m)}$ from its respective arm set ($\mathcal{K}$ in homogeneous settings and $\mathcal{K}^{(m)}$ in heterogeneous settings). The environment generates sub-Gaussian pre-attack reward feedback, denoted as $X_t^{(m,0)}(k_t^{(m)})$. We assume that there exists an *attacker* who chooses a subset of agents, $\mathcal{D} \subseteq \mathcal{M}$, as the target to attack.[1] It can observe pre-attack rewards from all agents, and manipulate those from $m \in \mathcal{D}$ into the post-attack reward $X_t^{(m)}(k_t^{(m)})$ before returning them to the agent. It is worth noting that the agents are oblivious to the attacker's presence and rely on this post-attack reward for decision-making. In

---

[1]For simplicity of presentation, we consider that all agents in $\mathcal{M}$ are potential targets. Results can be extended to scenarios where the attacker chooses $\mathcal{D} \subseteq \mathcal{M}_a \subseteq \mathcal{M}$, with $\mathcal{M}_a$ being the set of attackable agents.

homogeneous settings, analogous to attacks in single-agent scenarios, the attacker attempts to force *all* agents to pull a target arm for $T - o(T)$ times, incurring a cumulative attack cost of only

$$C(T) := \sum_{m \in \mathcal{D}} \sum_{t=1}^{T} \left| X_t^{(m,0)}(k_t^{(m)}) - X_t^{(m)}(k_t^{(m)}) \right| = o(T). \tag{1}$$

In heterogeneous settings, the attacker knows the local arm set $\mathcal{K}^{(m)}$ for every agent $m \in \mathcal{M}$, and its objective is to maximize the number of agents (affected agents) that suffer linear regret, as achieving the target arm objective with sublinear costs may not be feasible in this context. The detailed reason is discussed in Section 4.

## 3 Warm-up: Homogeneous Settings

CMA2B in homogeneous settings can be broadly classified into two categories (Wang et al., 2023a): fully distributed algorithms and leader-follower algorithms. The distinction between them lies in the presence or absence of a central agent (or server) that determines the actions of agents. In fully distributed algorithms, all agents participate in exploration and exploitation. Conversely, in leader-follower algorithms, the leader primarily manages exploration and plays a pivotal role. Due to the space limit, we focus on attacks against fully distributed algorithms in this section. We also design an attack strategy for the leader-follower algorithm DPE2 (Wang et al., 2020a) and provide a detailed cost analysis in Appendix B.4. Our analysis shows that targeting only the leader is adequate for misleading leader-follower algorithms.

**Target Algorithms**. There is an abundance of fully distributed algorithms in the literature; however, for consistency with the heterogeneous setting (Section 4), we study CO-UCB (Yang et al., 2022), a representative CMA2B algorithm that functions effectively in both homogeneous and heterogeneous environments, as our attack target. With CO-UCB, in each round, each agent pulls the arm with the highest UCB index and shares its reward observation immediately with others. We also extend our attack methodologies to UCB-TCOM (Wang et al., 2023a), which incorporates efficient communication mechanisms. Due to space constraints, we defer the detailed algorithm and analysis to Appendix B.3, where we address the issue of delayed feedback between agents. While most existing CMA2B algorithms are UCB-based, it is also possible to extend our findings to arbitrary no-regret CMA2B algorithms, based on the general attack strategy for single-agent bandits proposed by Liu & Shroff (2019). For further discussion, see Section 4.2.4.

Let $\hat{n}_t(k)$ denote the total number of times that arm $k$ is pulled by all $M$ agents globally up to time $t$. Without loss of generality, we choose the worst arm $K$ as the target arm, as it leads to the highest attack costs. Our goal is to mislead the agents running CO-UCB in order to convince them to pull the target arm $T - o(T)$ times with $o(T)$ attack costs. In the homogeneous setting, we can achieve this goal by merely attacking a single agent. Intuitively, since agents consistently share their reward observations, the manipulated rewards from one agent are disseminated to the rest, influencing their choices. To this end, we select an arbitrary agent, $m$, to attack. In round $t$, if its chosen arm $k$ is not the target arm $K$, we manipulate its current reward such that its updated empirical mean, after the attack, is below that of the target arm. The detailed attack value design can be found in Appendix B.1. This is inspired by (Jun et al., 2018), which studied attacks against single-agent bandits. However, the attack analysis in the multi-agent setting is more involved than that in the single-agent setting: it requires a carefully treatment on the delayed feedback between agents.

We define the reward mean gap of two arms as $\Delta(k, k') := \mu(k) - \mu(k')$. Theorem 1 provides the upper bound of the cumulative attack cost $C(T)$ of our attack strategy with parameters $\Delta_0$ and $\delta$ (note we only attack one agent) against CO-UCB with confidence parameter $\alpha$.

**Theorem 1.** *Suppose $T > K, \delta < 1/2, \Delta_0 > 0$. With probability $1 - \delta$, our attack strategy misguides* all agents *running CO-UCB to choose the target arm $K$ at least $T - o(T)$ times, or formally,*

$$\hat{n}_T(K) \geqslant MT - \frac{\alpha(K-1)}{2\Delta_0^2} \log T,$$

*using a cumulative cost at most*

$$C(T) \leqslant \left( \frac{\alpha}{2\Delta_0^2} \log T \right) \sum_{k < K} (\Delta(k, K) + \Delta_0) + \frac{4(K-1)\sigma}{\Delta_0} \sqrt{\alpha \log T \log \frac{K\pi^2 \alpha^2 (\log T)^2}{12\delta\Delta_0^4}}.$$

By setting $\Delta_0 = \Theta(\sqrt{\log T})$, the cumulative attack cost is bounded by $\hat{O}(K\sqrt{\log T})$, where $\hat{O}$ ignores $\log \log T$ factors. This matches the $\Omega(\sqrt{\log T})$ lower bound for the attack cost when targeting single-agent UCB as established in Zuo (2024). Moreover, it suggests that even when attacks are limited to a single agent, no additional costs are incurred, as the corrupted observations would propagate to other agents. Intuitively, it is equivalent to evenly spreading $\hat{O}(K/M\sqrt{\log T})$ costs to each agent. Notably, the total cost is independent of the number of agents $M$, which highlights the vulnerabilities in `CMA2B`, as the cost does not escalate with an increase in the number of agents. We also provide cost analyses when $T$ is unknown in Appendix B.2, adapting single-agent attack strategies from Zuo (2024).

## 4 ATTACKS IN HETEROGENEOUS SETTINGS

In this section, we study adversarial attacks on `CMA2B` in heterogeneous settings, where agents may have distinct local arm sets. We focus on the CO-UCB algorithm (Yang et al., 2022), a representative `CMA2B` algorithm for heterogeneous settings. While there are other heterogeneous algorithms (Wang et al., 2023b; Baek & Farias, 2021), all existing algorithms rely on UCB. Therefore, we choose CO-UCB as our example for devising attack strategies and believe our methodology can be extended to other UCB-based heterogeneous algorithms. We first discuss the viability of different attack objectives. Following that, we propose attack strategies with the appropriate objective and offer theoretical analyses of their associated costs.

### 4.1 ATTACK OBJECTIVES

While the majority of heterogeneous `CMA2B` algorithms are derivatives of their homogeneous counterparts, the distinctiveness introduced by agent heterogeneity poses novel challenges in devising adversarial attacks. We first consider the original target arm attack as in the homogeneous settings, which aims to deceive all agents into selecting a target arm $T - o(T)$ times. Intriguingly, in heterogeneous settings, achieving this objective might require linear attack costs. A simple example of two agents is shown in Figure 1a. We consider the target arm to be arm 3 in agent 1, and intuitively, once arm 1 or 2 is pulled, the attacker needs to decrease their rewards. However, given this heterogeneous setup, agent 2 only has access to arms 1 and 2. Therefore, it is compelled to select them repeatedly, and their reward samples are subsequently sent to agent 1. As a result, to deceive agent 1 into frequently selecting arm 3, linear attack costs on agent 2 become necessary. Proposition 1 formally shows the necessity of linear costs to realize the target arm attack.

**Proposition 1.** *For any attack strategy that can mislead the agents running CO-UCB in Figure 1a to pull the target arm $T - o(T)$ times, its attack cost is at least $C(T) \geqslant cT$ for some constant $c > 0$.*

Thus, the target arm attack may not be an appropriate attack objective in heterogeneous environments. Shifting our focus, we consider an alternative attack objective: misleading all agents toward linear regrets. While this objective seems less stringent, as it merely mandates agents not to choose their local optimal arms, the intrinsic heterogeneity of available arms brings forth complexities. Notably, there might be agents that, given the disparity in arm sets, cannot be simultaneously misguided towards linear regrets with only sublinear costs. Such agents are termed as *"conflict"* agents. An example of this scenario is depicted in Figure 1b. Our objective is to deceive agents 1 and 2, preventing them from selecting their locally optimal arms, and thereby incurring linear regrets. Notably, while arm 2 is suboptimal for agent 1, it is optimal for agent 2. After the attacks, agent 1 should pull arm 2 linear times, and those rewards will be communicated to agent 2 to affect its arm selection. Consequently, ensuring agent 2 chooses its suboptimal arm 3 almost linear times incurs linear attack costs. In Proposition 2, we demonstrate that linear costs are inevitable when pursuing this objective.

**Proposition 2.** *For any attack strategy that can successfully mislead all agents running CO-UCB in Figure 1b to suffer linear regrets, its attack cost is at least $C(T) \geqslant cT$ for some constant $c > 0$.*

Although sublinear attack costs might not be sufficient in leading all agents to experience linear regrets, they can still influence a subset of the agents. This realization prompts the final attack objective explored in this section: leveraging sublinear attack costs to misguide the maximum number of agents, aiming to increase the overall count of agents enduring linear regrets. To realize this goal, it is necessary to identify the largest set of agents that do not have conflicts. The condition under

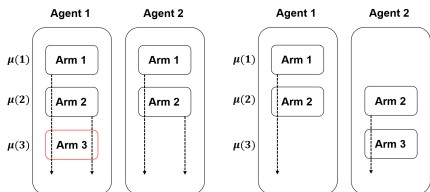
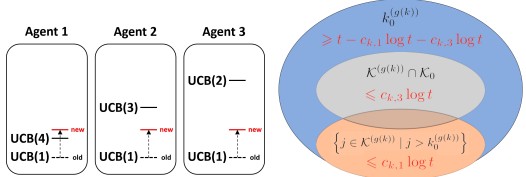

(a) Target arm attack  (b) Linear regret attack

Figure 1: Illustration of attack objectives

(a) Target agent selection.  (b) Actions of agent $g(k)$.

Figure 2: Proof ideas of Theorem 3.

---

**Algorithm 1** AAS: Affected Agents Selection

---

1: **input** local arm sets $\mathcal{K}^{(m)}$, $\forall m \in \mathcal{M}$
2: **initialize** $\mathcal{D}_0 \leftarrow \emptyset$, $\mathcal{K}_0 \leftarrow \emptyset$
3: Classify all agents into $\{\mathcal{M}_*(1), \dots, \mathcal{M}_*(K)\}$ according to their local optimal arms
4: Sort these agent sets according to set size such that $\mathcal{M}_*(\omega(k))$ is the agent subset that contains the $k^{\text{th}}$ largest number of agents
5: **for** $k = 1, 2, \dots, K$ **do**
6:     **if** for all agent $m \in \mathcal{D}_0 \cup \mathcal{M}_*(\omega(k))$, we have $|\mathcal{K}^{(m)} \setminus (\mathcal{K}_0 \cup \{\omega(k)\})| > 0$ **then**
7:         $\mathcal{D}_0 \leftarrow \mathcal{D}_0 \cup \mathcal{M}_*(\omega(k))$, $\mathcal{K}_0 \leftarrow \mathcal{K}_0 \cup \{\omega(k)\}$
8: **return** $\mathcal{D}_0, \mathcal{K}_0$

---

which two agents are deemed to be in conflict (i.e., they cannot be simultaneously attacked with sublinear costs) is given in Condition 1.

**Condition 1** (Agents conflict). *For any agents $m, m' \in \mathcal{M}$, if $|\mathcal{K}^{(m)} \setminus \{k_*^{(m)}, k_*^{(m')}\}| = 0$ or $|\mathcal{K}^{(m')} \setminus \{k_*^{(m)}, k_*^{(m')}\}| = 0$, we say agent $m$ conflicts with agent $m'$.*

### 4.2 ORACLE ATTACK STRATEGY

We study a scenario in which the attack algorithm has prior knowledge of the environment. Specifically, the attacker knows the reward ranking of all arms. This premise is less stringent than the "oracle attack" assumption from previous studies (Jun et al., 2018; Liu & Shroff, 2019), which requires precise knowledge of the mean rewards for every arm. Note that while we similarly label our approach as an "oracle attack" and use the $\mu(k)$ notation in our algorithms, they only rely on the relative arm ranking.

#### 4.2.1 AFFECTED AGENTS SELECTION

As previously mentioned, the first step for the attacker is to identify the largest group of agents without conflicts as the attack objective. Although Condition 1 allows us to check the conflict status between any pair of agents, finding the maximal conflict-free agent group poses a non-trivial combinatorial optimization challenge. To solve this problem, we propose the Affected Agents Selection (AAS) algorithm, described in Algorithm 1. Initially, the algorithm categorizes all agents into separate sets based on their local optimal arms, i.e., agent set $\mathcal{M}_*(k) := \{m \in \mathcal{M} : k_*^{(m)} = k\}$ contains all agents whose local optimal arm is arm $k$, and sort them by set size $M_*(k) := |\mathcal{M}_*(k)|$. Following this, a greedy set selection process is employed. The group of selected agents is maintained by $\mathcal{D}_0$. The algorithm examines sets from $M_*(\omega(1))$ to $M_*(\omega(k))$ sequentially, where $M_*(\omega(i))$ is the size of the $i^{\text{th}}$ largest agent sets among $\{\mathcal{M}_*(1), \dots, \mathcal{M}_*(K)\}$. If all agents within $\mathcal{M}_*(\omega(k))$ do not conflict with the current group $\mathcal{D}_0$ (Line 6), they are incorporated into $\mathcal{D}_0$, and the subsequent set $\mathcal{M}_*(\omega(k+1))$ will be examined. It eventually outputs the target group $\mathcal{D}_0$ and the local optimal arms' set $\mathcal{K}_0$. The following theorem provides its theoretical guarantee.

**Theorem 2.** *Algorithm 1 finds a $(1-1/e)$-approximate solution, i.e., $|\mathcal{D}_0| \geqslant (1-1/e)D_{\max}$, where $D_{\max}$ is the size of the largest conflict-free agent group.*

#### 4.2.2 TARGET AGENTS SELECTION

With output agent group $\mathcal{D}_0$ as the attack objective, a natural question arises: is it feasible to attack this group using only a limited number of agents, similar to the case in homogeneous settings? To address this, we introduce the Target Agents Selection (TAS) algorithm, which refines the selection of target agents to attack within $\mathcal{D}_0$. As described in Algorithm 2, for each agent $m \in \mathcal{D}_0$, it finds $k_0^{(m)}$, the local arm with the highest mean reward excluding all arms in $\mathcal{K}_0$ (Lines 5-6). Then, for each arm $k \in \mathcal{K}_0$, it checks $\mathcal{D}_{0,*}(k)$, which contains all agents in $\mathcal{D}_0$ with local optimal arm $k$; it chooses the agent in $\mathcal{D}_{0,*}(k)$ with the lowest $\mu(k_0^{(m)})$ and include it into the target agent set $\mathcal{G}_0$. Intuitively, in order to attack each arm $k \in \mathcal{K}_0$, Algorithm 2 selects the agent in $\mathcal{D}_0$ that is most likely to pull arm $k$ very often. Such target agent selection will help us control the number of times that $k \in \mathcal{K}_0$ is pulled by agents outside $\mathcal{G}_0$, ensuring successful attacks.

---

**Algorithm 2** TAS: Target Agents Selection

1: **input** $\mathcal{D}_0, \mathcal{K}_0$
2: **initialize** $\mathcal{G}_0 \leftarrow \emptyset$
3: **for** $m \in \mathcal{D}_0$ **do** ▷ for each affected agent
4:      $\mathcal{K}_0^{(m)} \leftarrow \mathcal{K}^{(m)} \setminus \mathcal{K}_0$
5:      $k_0^{(m)} \leftarrow \arg\max_{k \in \mathcal{K}_0^{(m)}} \mu(k)$
6: **for** $k \in \mathcal{K}_0$ **do** ▷ for local optimal arms
7:      $g(k) \leftarrow \arg\min_{m \in \mathcal{D}_{0,*}(k)} \mu(k_0^{(m)})$
8:      $\mathcal{G}_0 \leftarrow \mathcal{G}_0 \cup \{g(k)\}$
9: **return** $\mathcal{G}_0$

---

**Algorithm 3** Oracle Attack

1: **input** $\Delta_0$
2: $\mathcal{D}_0, \mathcal{K}_0 \leftarrow \text{AAS}((\mathcal{K}^{(m)})_{m \in \mathcal{M}})$
3: $\mathcal{G}_0 \leftarrow \text{TAS}(\mathcal{D}_0, \mathcal{K}_0)$
4: **for** $t = 1, 2, \cdots,$ **do**
5:      **for** agent $m \in \mathcal{G}_0$ **do**
6:          **if** $k_t^{(m)} \in \mathcal{K}_0$ **then**
7:              Attack $k_t^{(m)}$ with Equation (2)

---

#### 4.2.3 ATTACK STRATEGY AND ANALYSIS

We present the Oracle Attack (OA) algorithm as detailed in Algorithm 3. Initially, it invokes both AAS and TAS to select the target agent set $\mathcal{G}_0$ responsible for executing the attacks. Subsequently, whenever an agent $m \in \mathcal{G}_0$ chooses an arm $k \in \mathcal{K}_0$, it attacks $k$ with attack value $\gamma_t^{(m)}(k)$ to satisfy the ensuing inequality:

$$\hat{\mu}_t(k) \leq \min_{k' \in \mathcal{K} \setminus \mathcal{K}_0} \{\hat{\mu}_{t-1}(k') - 2\beta(\hat{n}_{t-1}(k')) - \Delta_0\}, \tag{2}$$

where $\hat{\mu}_t(k) = \frac{\hat{\mu}_{t-1}(k)\hat{n}_{t-1}(k) + \sum_{m'=1}^{M} X_t^{(m',0)}(k) - \gamma_t^{(m)}(k)}{\hat{n}_t(k)}$ and $\beta(N) = \sqrt{\frac{2\sigma^2}{N} \log \frac{\pi^2 K N^2}{3\delta}}$.

We provide the attack cost analysis for Algorithm 3.

**Theorem 3.** *Suppose $T > T_0, \alpha > 2$, where $T_0$ is a time-independent constant fulfills Eq. equation 3. With probability at least $1 - \delta$, Algorithm 3 misguides the agents to suffer regret at least*

$$R(T) \geqslant \sum_{k \in \mathcal{K}_0} \left( M_*(k) \Delta(k, k+1) T - \frac{2\alpha \log T}{\Delta_0^2} \right),$$

*using the cumulative cost at most*

$$C(T) \leqslant \sum_{k \in \mathcal{K}_0} \left( \frac{\alpha \log T}{2\Delta_0^2}(\Delta(k, K) + \Delta_0) + T_0 + \frac{4\sigma}{\Delta_0} \sqrt{\alpha \log T \log \frac{K \pi^2 \alpha^2 (\log T)^2}{12\delta \Delta_0^4}} \right).$$

*$T_0$ is a feasible solution of the following equation*

$$\frac{t}{\log t} \geqslant \max_{k \in \mathcal{K}_0} c_k, \tag{3}$$

*where $c_k = c_{k,1} + c_{k,2} + c_{k,3}$, and $c_{k,1} = \sum_{k' \in \mathcal{K}_0^{(g(k))} \setminus \{k_0^{(g(k))}\}} \frac{\alpha}{2\Delta^2(k_0^{(g(k))}, k')}, c_{k,2} = \frac{\alpha}{2 \min_{m \in \mathcal{D}_{0,*}(k) : k_0^{(m)} \neq k_0^{(g(k))}} \Delta^2(k_0^{(m)}, k_0^{(g(k))})}, c_{k,3} = |\mathcal{K}^{(g(k))} \cap \mathcal{K}_0| \cdot \frac{\alpha}{\Delta_0^2}.$*

By taking the optimal $\Delta_0$, the attack cost follows the order of $\hat{O}(|\mathcal{K}_0|\sqrt{\log T})$ and is independent of the number of affected agents, $|\mathcal{D}_0|$. This implies that a small attack cost can have a substantial impact on numerous agents in heterogeneous settings. Notice that directly comparing this result with Theorem 1 would be unfair, given the distinct objectives and settings they address. As Theorem 3 is one of our main technical contributions, we briefly discuss some key ideas below.

*Understanding Equation* (3). The feasible solution of Equation (3), $T_0$, is a threshold, after which our chosen target agents will consistently be the first to pull the arms intended for attack, thereby ensuring sufficient attack opportunities. This event occurs if the best arm, excluding those to be attacked, denoted as $k_0^{(g(k))}$, has been pulled more than $c_{k,2} \log t$ times. We derive the lower bound for this (blue in Figure 2b) by subtracting from the total rounds $t$ the number of pulls of the attacked arms (grey in Figure 2b), $c_{k,3} \log t$, and the number of pulls of arms worse than $k_0^{(g(k))}$ (orange in Figure 2b), $c_{k,1} \log t$. Thus, when $t - c_{k,1} \log t - c_{k,3} \log t \geqslant c_{k,2} \log t$, the expected event providing sufficient attack opportunities occurs, enabling us to demonstrate the success of attacks with bounded attack costs.

*Proof Challenge*. The main challenge in the proof arises from our choice of target agents. These agents can lower the post-attack empirical means of target arms in $\mathcal{K}_0$ only when they pull these arms. However, non-target agents may also pull these arms, yielding non-attacked samples that increase the empirical means towards the true means. This issue is especially prominent in heterogeneous settings where target and non-target agents, due to their distinct arm sets, might choose different arms to pull. Conversely, in homogeneous settings, all agents have access to all arms, consistently offering opportunities for attacks. To address this, we use TAS to choose target agents from $\mathcal{D}_0$. These target agents are the most likely ones to frequently pull arms in $\mathcal{K}_0$ because their top arms (after excluding those in $\mathcal{K}_0$), denoted as $k_0^{(m)}$, possess the least attractive mean rewards. For example, in Figure 2a, we consider three agents with local arm sets $\{1, 4\}, \{1, 3\}, \{1, 2\}$. Our TAS algorithm chooses agent 1 as the target agent: as shown in the figure, whenever the UCB of arm 1 increases and exceeds the others, with high probability, agent 1 will be the first to pull arm 1 since it has the worst local suboptimal arm, providing sufficient attack opportunities.

*Proof Sketch*. The key step is to prove that for any $t > T_0, k \in \mathcal{K}_0$, our target agents in $\mathcal{G}_0$ will always be the first to pull arm $k$ before non-target agents, providing enough attack opportunities. We first find a sufficient condition of this: if $k_0^{(g(k))}$ has been sufficiently pulled, i.e., $\hat{n}_t(k_0^{(g(k))})$ (the blue part in Figure 2b) is larger than $c_{k,2} \log t$, then $g(k)$ will consistently be the first to pull arm $k$ in $\mathcal{D}_{0,*}(k)$. We then look for a lower bound of $\hat{n}_t(k_0^{(g(k))})$ to satisfy this condition. There are two cases where $g(k)$ does not pull $k_0^{(g(k))}$: $g(k)$ can pull local arms worse than $k_0^{(g(k))}$ at most $c_{k,1} \log t$ times (the orange part in Figure 2b), and local arms within $\mathcal{K}_0$ at most $c_{k,3} \log t$ times (the grey part in Figure 2b). Hence, we find the lower bound of $\hat{n}_t(k_0^{(g(k))}) \geqslant t - c_{k,1} \log t - c_{k,3} \log t$, and we want $t - c_{k,1} \log t - c_{k,3} \log t \geqslant c_{k,2} \log t$ to satisfy the sufficient condition. We need to ensure this equation for every $k$, leading to Equation (3). Since the right-hand side of Equation (3) is a problem-dependent coefficient, we can derive $T_0$ which is independent of $t$. For subsequent rounds $t > T_0$, it is easy to prove the successful attacks with bounded attack costs.

### 4.2.4 DISCUSSIONS

**Discussion on General Attack Strategies**. In this paper, we focus on representative UCB-based algorithms, including CO-UCB (Yang et al., 2022), UCB-TCOM (Wang et al., 2023a), and DPE2 (Wang et al., 2020a). However, our findings can be extended to more general `CMA2B` algorithms, i.e., arbitrary no-regret bandit algorithms. In homogeneous settings, by substituting our initial attack strategy on the single affected agent with the general attack strategy for single-agent bandits from Section 4.2 of Liu & Shroff (2019), we can conduct an attack cost analysis similar to that in Theorem 1 (see Appendix B.5 for more details). In heterogeneous settings, AAS and TAS remain effective methods for identifying agents vulnerable to attacks and finding target agents. Nevertheless, a more detailed investigation is required to analyze the attack cost when applying the general attack strategy to these target agents, as our current analysis in Theorem 3 depends on specific characteristics of UCB algorithms. We leave this as a promising direction for future research.

**Discussion on Defense Strategies**. In the context of single-agent stochastic bandits, Zuo (2024) claims that, if some bandit algorithm achieves regret $REG(T)$ in the absence of attacks, then there

exists a strong attacker—who first observes the learner's action and then alters their reward observation—with attack cost $\Theta(REG(T))$ that can make the learner suffer linear regret (Fact 1 in (Zuo, 2024)). This implies that no algorithm can achieve sublinear regret when facing strong attackers with adaptive attack cost. Nevertheless, it remains possible to defend against strong attackers if their attack cost is limited, although designing and analyzing effective defense algorithms remains an open problem. Concurrently, there exists a complementary line of research focuses on developing robust single-agent bandit algorithms against corruption (Lykouris et al., 2018; Gupta et al., 2019), where weak attackers can alter the reward realizations but cannot observe the learner's current action.

Our work focuses on multi-agent stochastic bandits, uncovering latent vulnerabilities that arise through collaboration. Our findings from the attacker's perspective can also offer valuable insights into designing defense algorithms against strong attackers with limited attack cost. For instance, employing techniques such as TAS and AAS can help identify vulnerable agents that significantly impact others. By enhancing the protection of these critical agents, we may reduce the number of agents affected, thereby mitigating the impact of adversarial attacks.

### 4.3 EXTENSION TO LEARNING-THEN-ATTACK

In this section, we further relax the oracle assumption that the reward ranking of all arms is unknown. We introduce the Learning-Then-Attack (LTA) algorithm, detailed in Appendix A.6. We assume the minimal mean reward gap is positive, i.e., $\Delta_{\min} = \min_{k \neq k'} |\mu(k) - \mu(k')| > 0$, which was also adopted in prior multi-agent bandits studies (e.g., (Rosenski et al., 2016; Wang et al., 2020a)). LTA operates in two phases: an initial learning stage, where it discerns reward means and arm rankings, followed by an attack phase akin to Algorithm 3. In the learning stage, the arm ranking is pivotal for executing AAS and TAS, as they necessitate the knowledge of each agent's local optimal arms to optimize the affected agent group and minimize the target agent subset. To acquire this ranking, the attacker needs to compel agents to pull each arm multiple times. This ensures a sufficient number of pre-attack samples, allowing for a clear distinction between the LCBs and UCBs for every arm pair.

Given that agents choose arms based on UCB algorithms, the attacker needs to stimulate agents to collect ample samples for suboptimal arms by increasing their UCB values through attacks. More specifically, if the attacker wants to accumulate samples of arm $k$, it needs to attack the arm's reward such that the subsequent condition is met:

$$\mathrm{UCB}_t(k) > \mathrm{UCB}_t(k'), \ \forall k \neq k'.$$

Once the number of arm pulls for $k$ reaches a threshold $L := \lceil \frac{2\log(2K/\delta)}{\Delta_{\min}^2} \rceil$, the attacker resets the arm's empirical mean (remove prior attacks on the arm). This ensures that arms that have not been sufficiently sampled will be selected later. We introduce the following condition to discuss the number of target agents required for the learning stage.

**Condition 2** (Arm accessibility). *For each arm $k \in \mathcal{K}$, there are at least $cM \in \mathbb{N}^+$ agents in the target agent set $\mathcal{S}_0 \subseteq \mathcal{M}$ being able to access it, where $c > 0$ is the arm accessible rate among target agents. Formally,*

$$\left| \{m \in \mathcal{S}_0 : k \in \mathcal{K}^{(m)}\} \right| \geqslant cM, \quad \forall k \in \mathcal{K}.$$

We note that target agent set $\mathcal{S}_0$ for learning can be different from $\mathcal{G}_0$ chosen in Algorithm 2, and this condition is not restrictive. For example, letting $\mathcal{S}_0$ be a subset of agents whose local optimal arms together cover the full arm set $\mathcal{S}$, choosing $c = \frac{1}{M}$ is always valid. Condition 2 ensures that during each round of the learning stage, there are at least $cM$ *effective* observations. Here, "effective observations" denote the observations of arms with sampling times below the threshold $L$. During this stage, agents are motivated to select arms with observations fewer than $L$. Whenever there is an arm that hasn't reached this threshold, a minimum of $cM$ agents will be motivated to select it. Consequently, the learning stage ends after no more than $\frac{KL}{cM}$ rounds.

**Analysis**. Upon completing the learning stage, the attacker has accurate estimates of the reward means for all arms to determine the arm ranking. It can then apply the oracle attack in Algorithm 3. Notice that in the initial phase of the second stage, when the first time that any arm in $\mathcal{K}_0$ is pulled by a target agent, the attacker incurs a significant attack cost. This is because the attacker needs to pay additional costs to alter the unbiased empirical means derived from the learning stage. However,

such extra costs can be upper bounded by $\frac{KL(\Delta(1,K)+\beta(1)+b)}{c}$, where $b$ is the upper bound of the mean rewards, i.e., $\mu(k) \leqslant b$. Subsequent to this adjustment, the OA algorithm operates identically to its behavior in the oracle setting.

**Theorem 4.** *Suppose $T > T_0, \alpha > 2, \delta < 0.5$, where $T_0$ is a time-independent constant fulfills Eq. equation 3. With probability at least $1 - 2\delta$, Algorithm 4 misguides the agents to suffer regret at least*

$$R(T) \geqslant \sum_{k \in \mathcal{K}_0} \left( M_*(k)\Delta(k, k+1)T - \frac{2\alpha \log T}{\Delta_0^2} \right),$$

*using the cumulative cost at most*

$$C(T) \leqslant \frac{4K(\Delta(1,K)+\beta(1)+b)\log T}{c\Delta_{\min}^2} + \sum_{k \in \mathcal{K}_0} \left( \frac{\alpha \log T}{2\Delta_0^2}(\Delta(k,K) + \Delta_0) + T_0 + \frac{4\sigma}{\Delta_0}\sqrt{\alpha \log T \log \frac{K\pi^2\alpha^2(\log T)^2}{12\delta\Delta_0^4}} \right).$$

Compared with the result of the oracle attack, the first term in the attack cost arises from the attacks during the rank learning stage, while the second term is the same as that in Theorem 3.

## 5 EXPERIMENTS

We conduct experiments in both homogeneous and heterogeneous settings. Due to space limitations, we only present the results of heterogeneous settings here; the results of homogeneous settings can be found in Appendix B.6. We take $T = 100,000, K = 20, M = 20$. The mean rewards of the arms are randomly sampled within $(0, 5)$ while ensuring $\Delta_{\min}^2 \geqslant 0.01$, and the reward of each arm $k$ follows the Gaussian distribution $\mathcal{N}\left( \mu(k), \sigma^2 \right)$ with $\sigma = 0.1$. Furthermore, each agent $m$

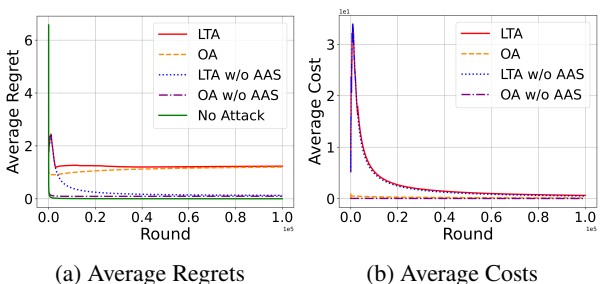

(a) Average Regrets      (b) Average Costs

Figure 3: Attacks against CO-UCB.

has a set of arms with $|\mathcal{K}^{(m)}| = 5$. The CO-UCB algorithm takes $\alpha = 10$, and the attack parameters are set to $\Delta_0 = 0.05$ and $\delta = 0.1$. We conducted experiments with five algorithms for comparison: Oracle Attack (OA) with and without Affected Agents Selection (AAS), Leaning-Then-Attack (LTA) with and without AAS, and No Attack. Each experiment was repeated 10 times. Figure 3a shows the average regret across various algorithms. Both LTA and OA result in the most significant average regrets, primarily due to AAS's capability to identify the most extensive group of affected agents. Without AAS, their average regrets converge to reduced constant values, indicating linear regrets for a limited subset of affected agents. Figure 3b shows the average attack costs of different algorithms. All of them approach zero, indicating sublinear cumulative attack costs. In particular, LTA initially incurs higher costs compared to OA, a consequence of the high attack costs during their learning stage.

## 6 CONCLUDING REMARKS

This paper explores adversarial attacks on `CMA2B`, revealing significant vulnerabilities in both homogeneous and heterogeneous settings. Our proposed attack strategies demonstrate how minimal manipulation of selected agents can degrade the performance of an entire cooperative learning system. It also opens up multiple future directions, including the development of attack strategies for competitive multi-agent bandits involving collisions. Further exploration could extend to dynamic and asynchronous systems, where agent populations and environments evolve over time. Finally, our insights into current `CMA2B` vulnerabilities highlight the need for designing robust algorithms that can withstand adversarial manipulation. Future research should focus on creating defenses that detect and mitigate attacks, ensuring the reliability of cooperative multi-agent systems in practice.

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

APPENDIX

## A    PROOFS

### A.1    PROOF OF THEOREM 1

*Proof.* We first define the "good event" $E = \{\forall k, \forall t > K : |\hat{\mu}_t^{(0)}(k) - \mu(k)| < \beta(\hat{n}_t(k))\}$, where $\hat{\mu}_t^0(k)$ is the pre-attack empirical mean of arm $k$ for all agents up to time slot $t$. By Hoeffding's inequality, we can prove that for any $\delta \in (0, 1)$, $P(E) > 1 - \delta$. To prove Theorem 1, we introduce two lemmas.

**Lemma 1.** *Assume event $E$ holds and $\delta \leqslant 1/2$. For any $k \neq K$ and any $t \geqslant K$, we have*

$$\hat{n}_t(k) \leqslant \min\{\hat{n}_t(K), \frac{\alpha \log t}{2\Delta_0^2}\} \tag{4}$$

*Proof.* Fix some $t > K$, we assume that $k_t^{(m)} = k \neq K$. Note that when $k_t^{(m)} \neq k$, $\hat{n}_t(k)$ will never increase. Also, we assume the last time arm $k$ is selected before time slot $t$ is $t'$. By our attack design in Equation (30), we have:

$$\hat{\mu}_{t'}(k) \leqslant \hat{\mu}_{t'}(K) - 2\beta(\hat{n}_{t'}(K)) - \Delta_0. \tag{5}$$

On the other hand, arm $k$ is selected in round $t$ means that it has a higher UCB than arm $K$ in round $t - 1$:

$$\hat{\mu}_{t-1}(k) + \sqrt{\frac{\alpha \log(t-1))}{2\hat{n}_{t-1}(k)}} \geqslant \hat{\mu}_{t-1}(K) + \sqrt{\frac{\alpha \log(t-1)}{2\hat{n}_{t-1}(K)}}. \tag{6}$$

Note that $\hat{\mu}_{t'}(k) = \hat{\mu}_{t-1}(k)$. Substituting Equation (5) into Equation (6), we have:

$$\begin{aligned}
\sqrt{\frac{\alpha \log(t-1))}{2\hat{n}_{t-1}(k)}} - \sqrt{\frac{\alpha \log(t-1)}{2\hat{n}_{t-1}(K)}} &\geqslant \hat{\mu}_{t-1}(K) - \hat{\mu}_{t-1}(k) \\
&\geqslant \hat{\mu}_{t-1}(K) - (\hat{\mu}_{t'}(K) - 2\beta(\hat{n}_{t'}(K)) - \Delta_0) \\
&\geqslant \Delta_0 \\
&> 0,
\end{aligned} \tag{7}$$

where the third inequality is due to event $E$ and the monotonically decreasing property of $\beta$. Therefore, by Equation (7) and $\sqrt{\frac{\alpha \log(t-1)}{2\hat{n}_{t-1}(K)}} > 0$, the proof is done. □

**Lemma 2.** *Assume event $E$ holds and $\delta \leqslant 1/2$. Denote $\tau(t, k)$ as the rounds in which arm $k$ is selected by all agents up to round $t$. For any $i \neq K$ and any $t \geqslant K$, we have*

$$\sum_{s \in \tau(t,k)} \gamma^{(m)}(s) \leqslant \hat{n}_t(k)(\Delta(k, K) + \Delta_0 + 3\beta(\hat{n}_t(K)) + \beta(\hat{n}_t(k))) \tag{8}$$

*Proof.* We assume that $k_t^{(m)} = k \neq K$ for some fixed $t > K$. By Equation (30), we can compute the attack value in each round $t$:

$$\begin{aligned}
\gamma^{(m)}(t) &= \left( \hat{\mu}_{t-1}(k)\hat{n}_{t-1}(k) + \sum_{m'=1}^{M} X_t^{(m',0)}(k) - \hat{n}_t(k)(\hat{\mu}_t(K) - 2\beta(\hat{n}_t(K)) - \Delta_0) \right)_+ \\
&= \left( \hat{\mu}_{t-1}^{(0)}(k)\hat{n}_{t-1}(k) + \sum_{m'=1}^{M} X_t^{(m',0)}(k) - \sum_{s \in \tau(t-1,k)} \gamma^{(m)}(s) - \hat{n}_t(k)(\hat{\mu}_t(K) - 2\beta(\hat{n}_t(K)) - \Delta_0) \right)_+.
\end{aligned} \tag{9}$$

If $\gamma^{(m)}(t) = 0$, we change to examine the last time it was greater than zero. We have

$$\sum_{s \in \tau(t,k)} \gamma^{(m)}(s) = \hat{\mu}_{t-1}^{(0)}(k)\hat{n}_{t-1}(k) + \sum_{m'=1}^{M} X_t^{(m',0)}(k) - \hat{n}_t(k)(\hat{\mu}_t(K) - 2\beta(\hat{n}_t(K)) - \Delta_0)$$

$$= \hat{n}_t(k)(\hat{\mu}_t^{(0)}(k) + \hat{\mu}_t(K) + 2\beta(\hat{n}_t(K)) + \Delta_0)$$
$$\leqslant \hat{n}_t(k)(\Delta(k,K) + \Delta_0 + 3\beta(\hat{n}_t(K)) + \beta(\hat{n}_t(k))),$$

(10)

where the last inequality is due to event $E$. $\qquad\square$

With Lemma 1, we can easily get that the target arm $K$ is selected for at least $MT - (K - 1)(\frac{\alpha}{2\Delta_0^2} \log T)$ times. As for the cumulative cost, we sum over all non-target arms using Lemma 2. We have:

$$\sum_t^T \gamma^{(m)}(t) \leqslant \sum_{k=1}^{K-1} \hat{n}_t(k)(\Delta(k,K) + \Delta_0) + 4 \sum_{k=1}^{K-1} \hat{n}_t(k)\beta(\hat{n}_t(k))$$

$$\leqslant \left(\frac{\alpha}{2\Delta_0^2} \log T\right) \sum_{k<K} (\Delta(k,K) + \Delta_0) + \frac{4(K-1)\sigma}{\Delta_0} \sqrt{\log T \log \frac{K\pi^2\alpha^2(\log T)^2}{12\delta\Delta_0^4}},$$

(11)

where in the last inequality, we substitute the chosen times of each arm $k$, $\hat{n}_t(k)$, into $\beta$. $\qquad\square$

### A.2 Proof of Proposition 1

The proofs of Proposition 1 and Proposition 2 are inspired by the proof of Theorem 2 in Zuo (2020).

*Proof.* Assume arm 3 is our target arm. We first introduce some notations. Let $\gamma(t,m,k) = X_t^{(m,0)}(k) - X_t^{(m)}(k)$. Note that if $k_t^{(m)} \neq k$, then $\gamma(t,m,k) = 0$. In addition, let $\Gamma(t,k) = \sum_{s=1}^t \sum_{m=1}^2 |\gamma(s,m,k)|$.

Assume event $E$ holds. Consider the last time that arm 3 is pulled (by agent 1) is round $t + 1$. As the agent selects the arm with the highest UCB in time slot $t$, we have:

$$\hat{\mu}_t(3) + \sqrt{\frac{\alpha \log t}{2\hat{n}_t(3)}} \geqslant \hat{\mu}_t(k) + \sqrt{\frac{\alpha \log t}{2\hat{n}_t(k)}},$$

(12)

for $k = 1, 2$. Also, we have the following two inequalities:

$$\frac{\hat{\mu}_t(3)\hat{n}_t(3) - \Gamma(t,3)}{\hat{n}_t(3)} \leqslant \mu(3) + \beta(\hat{n}_t(3)),$$

(13)

$$\frac{\hat{\mu}_t(k)\hat{n}_t(k) + \Gamma(t,k)}{\hat{n}_t(k)} \geqslant \mu(k) - \beta(\hat{n}_t(k)),$$

(14)

for $k = 1, 2$. This is because we consider the absolute values of both positive and negative instances of $\gamma$ up to round $t$ and also utilize event $E$ to upper and lower bound the empirical means. Note that arm 3 can only be selected by agent 1. Therefore, we have $\hat{n}_t(3) \leqslant t$, and $\max\{\hat{n}_t(1), \hat{n}_t(2)\} \geqslant t/2$. Let $i = \arg\max_{k=1,2}\{\hat{n}_t(k)\}$. Then, we can construct an instance that needs linear cumulative cost to pull the target arm for linear times.

Suppose $\Delta(2,3) \geqslant 4, \sigma = 0.5$. As $\hat{n}_T(3) = T - o(T)$, we can assume that there exists some constant $\frac{\sqrt{\delta}}{\pi} < c < 1$, which satisfies $\min\{\hat{n}_t(i), \hat{n}_t(3)\} \geqslant ct$. Putting Equations (12) to (14) together, we have:

$$\frac{\Gamma(t,3)}{\hat{n}_t(3)} + \frac{\Gamma(t,i)}{\hat{n}_t(i)} \geqslant \Delta(i,3) - \sqrt{\frac{\alpha \log t}{2\hat{n}_t(3)}} - \beta(\hat{n}_t(3)) - \beta(\hat{n}_t(i))$$

$$\geqslant \Delta(2,3) - 3\beta(ct)$$

$$> \Delta(2,3) - 3\sqrt{\frac{\delta}{\pi t} \log \frac{\pi t}{\delta}}$$

$$> 1,$$

(15)

where the second inequality holds because $\Delta(i,3) \geqslant \Delta(2,3)$, and we have $\sqrt{\frac{\alpha \log t}{2\hat{n}_t(3)}} \leqslant \beta(\min\{\hat{n}_t(i), \hat{n}_t(3)\}) \leqslant \beta(ct)$ when $\frac{\sqrt{\delta}}{\pi} < c < 1$, as $\beta$ is monotonically decreasing. The third inequality can be derived from the following result:

$$\beta(ct) = \sqrt{\frac{1}{2ct} \log \frac{\pi^2 c^2 t^2}{\delta}}$$
$$\overset{c<1<\frac{1}{\sqrt{\delta}}}{\leqslant} \sqrt{\frac{1}{2ct} \log(\frac{\pi t}{\delta})^2}$$
$$\overset{c>\frac{\sqrt{\delta}}{\pi}>\frac{\delta}{\pi}}{\leqslant} \sqrt{\frac{\delta}{\pi t} \log \frac{\pi t}{\delta}},$$

and the last inequality in Equation (15) is due to $\sqrt{\frac{\log x}{x}} < 1$ for any $x \geqslant 1$. Therefore, the cumulative cost is

$$C(T) \geqslant \Gamma(T,3) + \Gamma(T,i) \geqslant cT. \tag{16}$$

$\square$

### A.3 PROOF OF PROPOSITION 2

*Proof.* We use the same notations in Appendix A.2 and assume event $E$ holds. In this scenario, we assume that both arms 2 and 3 should be selected for linear times; otherwise, one of these agents will not suffer linear regret.

Consider the last time arm 3 is pulled (by agent 2) is $t + 1$. Then we have the UCB order in time slot $t$:

$$\hat{\mu}_t(3) + \sqrt{\frac{\alpha \log t}{2\hat{n}_t(3)}} \geqslant \hat{\mu}_t(2) + \sqrt{\frac{\alpha \log t}{2\hat{n}_t(2)}}. \tag{17}$$

Also, we have the following two inequalities for the same reason of Equations (13) and (14):

$$\frac{\hat{\mu}_t(3)\hat{n}_t(3) - \Gamma(t,3)}{\hat{n}_t(3)} \leqslant \mu(3) + \beta(\hat{n}_t(3)), \tag{18}$$

$$\frac{\hat{\mu}_t(2)\hat{n}_t(2) + \Gamma(t,2)}{\hat{n}_t(2)} \geqslant \mu(2) - \beta(\hat{n}_t(2)). \tag{19}$$

Suppose $\Delta(2,3) \geqslant 4, \sigma = 0.5$. As $\hat{n}_T(k) = T - o(T), k = 2,3$, we can assume that there exists some constant $\frac{\sqrt{\delta}}{\pi} < c < 1$, which satisfies $\min\{\hat{n}_t(2), \hat{n}_t(3)\} \geqslant ct$. Put Equations (17) to (19) together, we have:

$$\frac{\Gamma(t,3)}{\hat{n}_t(3)} + \frac{\Gamma(t,2)}{\hat{n}_t(2)} \geqslant \Delta(2,3) - \sqrt{\frac{\alpha \log t}{2\hat{n}_t(3)}} - \beta(\hat{n}_t(3)) - \beta(\hat{n}_t(2))$$
$$\geqslant \Delta(2,3) - 3\beta(ct)$$
$$> \Delta(2,3) - 3\sqrt{\frac{\delta}{\pi t} \log \frac{\pi t}{\delta}}$$
$$> 1,$$
$$\tag{20}$$

with the reason similar to that for Equation (15). Therefore, the cumulative cost is

$$C(T) \geqslant \Gamma(T,3) + \Gamma(T,2) \geqslant cT. \tag{21}$$

$\square$

### A.4 PROOF OF THEOREM 2

*Proof.* We define a set function $f : 2^\Omega \mapsto \mathbb{R}$, which takes the power set of $\Omega := \{\mathcal{M}_*(1), \cdots, \mathcal{M}_*(K)\}$ as input and outputs the largest number of conflict-free agents. We then check its submodularity. For every $X, Y \subseteq \Omega$ with $X \subseteq Y$ and every $x \in \Omega \setminus Y$, we have

$$f(X \cup \{x\}) - f(X) \geqslant f(Y \cup \{x\}) - f(Y), \tag{22}$$

since if agent $m \in x$ conflicts with any agent $m' \in X$, it must conflict with $m' \in Y$ as well. As a result, $f$ is submodular and the greedy algorithm in Algorithm 1 gives a $(1 - 1/e)$-approximate solution.

$\square$

## A.5 Proof of Theorem 3

*Proof.* For each arm $k \in \mathcal{K}_0$, we consider the target agent $g(k)$, which is mainly responsible for attacking $k$. We first prove that if $k_0^{(g(k))}$ has been sufficiently pulled, i.e., $\hat{n}_t(k_0^{(g(k))}) \geqslant c_{k,2} \log t$, $g(k)$ will consistently be the first to pull arm $k$ in $\mathcal{D}_{0,*}(k)$. With $\hat{n}_t(k_0^{(g(k))}) \geqslant c_{k,2} \log t$, for every $m \in \mathcal{D}_{0,*}(k)$ such that $k_0^{(m)} \neq k_0^{(g(k))}$, we have

$$UCB_t(k_0^{(g(k))}) = \hat{\mu}_t(k_0^{(g(k))}) + \sqrt{\frac{\alpha \log t}{2\hat{n}_t(k_0^{(g(k))})}} \leqslant \mu(k_0^{(g(k))}) + \Delta(k_0^{(m)}, k_0^{(g(k))}) = \mu(k_0^{(m)}) \leqslant UCB_t(m). \tag{23}$$

Since $k_0^{(g(k))}, k_0^{(m)}$ are the local optimal arms excluding all arms in $\mathcal{K}_0$ and $UCB_t(k_0^{(g(k))})$ is always less or equal to $UCB_t(k_0^{(m)})$, target agent $g(k)$ will pull arm $k$ earlier than any agent $m$, assuring enough attack opportunities.

Next, we want to derive a lower bound of $\hat{n}_t(k_0^{(g(k))})$ for $g(k)$. There are two cases in which $g(k)$ does not pull $k_0^{(g(k))}$: it can pull arms either in $\mathcal{K}_0^{(g(k))} \setminus \{k_0^{(g(k))}\}$ or in $\mathcal{K}^{(g(k))} \cap \mathcal{K}_0$. We first consider the former case. Since $k_0^{(g(k))}$ is the optimal arm in $\mathcal{K}_0^{(g(k))}$, the number of pulls of these suboptimal arms can be bounded by

$$\sum_{k' \in \mathcal{K}_0^{(g(k))} \setminus \{k_0^{(g(k))}\}} n_t^{(g(k))}(k') \leqslant \sum_{k' \in \mathcal{K}_0^{(g(k))} \setminus \{k_0^{(g(k))}\}} \frac{\alpha \log t}{2\Delta^2(k_0^{(g(k))}, k')} = c_{k,1} \log t, \tag{24}$$

where $n_t^{(g(k))}(k')$ is the number of times that agent $g(k)$ pulls arm $k'$. The inequality comes from the upper bound of the suboptimal arm pulls for UCB algorithms. We then discuss the latter case in which $k' \in \mathcal{K}^{(g(k))} \cap \mathcal{K}_0$ is pulled. If $\hat{n}_t(k') \geqslant \frac{\alpha \log t}{2\Delta_0^2}$, for any non-target agent $m \notin \mathcal{G}_0$, there always exists $k'' \in \mathcal{K}^{(m)} \setminus \mathcal{K}_0$ such that

$$UCB_t(k') = \hat{\mu}_t(k') + \sqrt{\frac{\alpha \log t}{2\hat{n}_t(k')}} \leqslant \mu(k'') - \Delta_0 + \sqrt{\frac{\alpha \log t}{2\hat{n}_t(k')}} \leqslant \mu(k'') \leqslant UCB_t(k''), \tag{25}$$

where the first inequality is due to our attack design in Equation (2). As a result, non-target agents will not pull $k'$ anymore. Equation (2) also ensures the number of pulls by target agents after $\hat{n}_t(k') \geqslant \frac{\alpha \log t}{2\Delta_0^2}$ is bounded by $\frac{\alpha \log t}{2\Delta_0^2}$ (similar to the proof of Lemma 1). Thus, the total number of arm pulls for all $k' \in \mathcal{K}^{(g(k))} \cap \mathcal{K}_0$ from agent $g(k)$ is upper bounded by $|\mathcal{K}^{(g(k))} \cap \mathcal{K}_0| \cdot \frac{2\alpha \log t}{2\Delta_0^2} = c_{k,3} \log t$. Then for every $k \in \mathcal{K}_0$, we want

$$\hat{n}_t(k_0^{(g(k))}) \geqslant t - c_{k,1} \log t - c_{k,3} \log t \geqslant c_{k,1} \log t. \tag{26}$$

With a feasible $T_0$ such that

$$\frac{T_0}{\log T_0} \geqslant \max_{k \in \mathcal{K}_0} c_k, \tag{27}$$

for any $t > T_0$, arms in $\mathcal{K}_0$ will only be pulled by target agents in $\mathcal{G}_0$, and these agents will conduct attacks according to Equation (2). When $t > T_0$, for every $k \in \mathcal{K}_0$, we can follow the same steps in the proof of Lemma 1 and obtain

$$\hat{n}_t(k) \leqslant \frac{\alpha \log t}{2\Delta_0^2}. \tag{28}$$

Based on this, the cost upper bound in Theorem 3 can be easily derived by following the same steps in the proof of Lemma 2, which concludes the proof. $\square$

---

**Algorithm 4** LTA: Learning-Then-Attack

---

*Proof.*  1: Input: confidence parameter $\delta$, minimal mean difference $\Delta_{\min}$, threshold $L$

    ▷ Stage 1:  Attack to learn full rank

2: **while** $\min_{k \in \mathcal{K}} \hat{n}_t(k) < L$ **do**

3:     **for all** agent $m \in \mathcal{M}$ **do**

4:         Observe the pulled arm $k_t^{(m)}$

5:         **if** $\hat{n}_t(k_t^{(m)}) < L$ **then**

6:             Attack $k_t^{(m)}$ according to Eq. equation 4

7:         **if** $\hat{n}_t(k_t^{(m)}) = L$ **then**

8:             Recover $k_t^{(m)}$ to unbiased mean

    ▷ Stage 2:  Attack to mislead agents

9: Run oracle attack in Algorithm 3

---

### A.6 PROOF OF THEOREM 4

We first prove that at the end of the learning stage, Algorithm 4 can learn the correct mean reward ranking of all arms. For every arm $k \in \mathcal{K}$, we have $\hat{n}_t(k) \geqslant L = \lceil \frac{2 \log(2K/\delta)}{\Delta_{\min}^2} \rceil$. By Hoeffding's inequality,

$$|\hat{\mu}_t(k) - \mu(k)| \leqslant \frac{\Delta_{\min}}{2} \tag{29}$$

with probability $1 - \delta$, which indicates that sorting all arms according to $\hat{\mu}_t(k)$ will give the correct ranking.

We then consider the attack cost incurred during the learning stage. As discussed in Section 4.3, the learning stage ends after no more than $\frac{KL}{cM}$ rounds. Since the attack value per round is upper bound by $\Delta(1, K) + \beta(1) + b$, the total attack cost of $M$ agents is bounded by $\frac{MKL(\Delta(1,K)+\beta(1)+b)}{cM}$, where $b$ is the upper bound of the mean rewards. In addition, Algorithm 4 also needs to pay a significant cost $\gamma_t^{(m)}$ to alter the unbiased empirical mean of $k \in \mathcal{K}_0$ for the first time that $k$ is attacked by $m \in \mathcal{G}_0$ during the second stage. This is due to the large value of $\hat{n}_{t-1}(k)$ when calculating $\hat{\mu}_t(k)$, which necessities a relatively large $\gamma_t^{(m)}$ to ensure Equation (2). Since this cost can still be upper bounded by $\frac{KL(\Delta(1,K)+\beta(1)+b)}{c}$, the total additional cost induced by the learning stage is $\frac{2KL(\Delta(1,K)+\beta(1)+b)}{c}$, which appears as the first term of $C(T)$ in Theorem 4. Notice that the learning stage of Algorithm 4 directly ensures that $k_0^{(g(k))}$ has been sufficiently pulled for every $k \in \mathcal{K}_0$. Thus, the cost of the attack stage is the same as that in Algorithm 3, which concludes the proof.

$\square$

## B ADDITIONAL RESULTS IN HOMOGENEOUS SETTINGS

### B.1 ATTACKS AGAINST CO-UCB

In this section, we give some details about the design of the algorithm in Section 3.

Our goal is to mislead the agents running CO-UCB in order to convince them to pull the target arm $T - o(T)$ times with $o(T)$ attack costs. In the homogeneous setting, we can achieve this goal by merely attacking a single agent. Intuitively, since agents consistently share their reward observations, the manipulated rewards from one agent are disseminated to the rest, influencing their choices. To this end, we select an arbitrary agent, $m$, to attack. In round $t$, if its chosen arm $k$ is not the target arm $K$, we manipulate its reward $X_t^{(m,0)}(k)$ to fulfill the following inequality:

$$\hat{\mu}_t(k) \leqslant \hat{\mu}_t(K) - 2\beta(\hat{n}_t(K)) - \Delta_0, \tag{30}$$

where

$$\hat{\mu}_t(k) = \frac{\hat{\mu}_{t-1}(k)\hat{n}_{t-1}(k) + \sum_{s=1}^{M} X_t^{(s,0)}(k) - \gamma^{(m)}(t)}{\hat{n}_t(k)},$$

$$\beta(N) := \sqrt{\frac{2\sigma^2}{N} \log \frac{\pi^2 K N^2}{3\delta}},$$

and $\gamma^{(m)}(t)$ is the attack value, $\Delta_0 > 0$ and $\delta > 0$ are the parameters of the attack strategy. This strategy is similar to the attack design against single-agent bandits (Jun et al., 2018). It guarantees that the empirical means of non-target arms, after the attack, consistently remain below that of the target arm. We define the reward mean gap of two arms as $\Delta(k, k') := \mu(k) - \mu(k')$. Theorem 1 provides the upper bound of the cumulative cost $C(T) = \sum_{t=1}^{T} |\gamma^{(m)}(t)|$ (note we only attack one agent $m$) for CO-UCB with confidence parameter $\alpha$. To make sure that Equation (30) always holds after attack, the $\gamma^{(m)}(t)$ can be computed as the following equality:

$$\gamma^{(m)}(t) = (\hat{\mu}_{t-1}(k)\hat{n}_{t-1}(k) + \sum_{s=1}^{M} X_t^{(s,0)}(k) - \hat{n}_t(k)(\hat{\mu}_t(K) - 2\beta(\hat{n}_t(K)) - \Delta_0))_+. \quad (31)$$

### B.2 ATTACKS AGAINST CO-UCB WITH UNKNOWN TIME HORIZON

In Section 3, we analyze the cost on attacking CO-UCB algorithm, and mention that the careful selection of $\Delta_0$ to $\Theta(\sqrt{\log T})$ can upper bound the cumulative cost by $\hat{O}(K\sqrt{\log T})$. However, this selection requires the attacker to know the time horizon $T$ in advance. In this section, we give an attack technique from Zuo (2024) which achieves the similar cost upper bound, but does not need the background of $T$ in advance.

We still only need to attack one agent, $m$. In round $t$, if arm $k \neq K$ is chosen, we attack agent $m$. Instead of Equation (30), our goal is changed to:

$$\hat{\mu}_t(k) = \hat{\mu}_t(K) - 2\beta(\hat{n}_t(K)) - \sqrt{\alpha} \exp(\hat{n}_t(k)), \quad (32)$$

where $\alpha$ is the confidence parameter of CO-UCB algorithm. Now, we first give the result, and then analyze it.

**Theorem 5.** *Suppose $T > K, \delta < 1/2$. With probability at least $1 - \delta$, the attack strategy, following Equation* (32)*, misguides all agents, running the CO-UCB algorithm, to choose the target arm $K$ at least $T - o(T)$ times, or formally,*

$$\hat{n}_T(K) \geqslant MT - (K-1)\log\log T,$$

*using a cumulative cost at most*

$$C(T) \leqslant \sum_{k<K} \log\log T(\Delta(k, K) + \exp\sqrt{\alpha \log T})$$

$$+ \sigma(K-1)\sqrt{32\log\log T \log \frac{\pi^2 K (\log\log T)^2}{3\delta}}.$$

*Proof.* Assume event $E$ holds throughout this section.

**Lemma 3.** *Assume event $E$ (the one we used before) holds. In any round $t > K$, $\hat{n}_t(k) \leqslant \lceil 0.5 \log\log t \rceil$ for any $k \neq K$.*

*Proof.* Suppose it is not true. Then, some non-target arm $k$ is pulled for more than $\lceil 0.5 \log\log t \rceil$ times in round $t$. Thus, we assume $\hat{n}_{t_0-1}(k) < \lceil 0.5 \log\log t \rceil \leqslant \hat{n}_{t_0}(k)$ for some $t_0 < t$. Then, we have

$$\hat{\mu}_{t_0}(k) \leqslant \hat{\mu}_{t_0}(K) - 2\beta(\hat{n}_{t_0}(K)) - \sqrt{\alpha} \exp(\hat{n}_t(k))$$
$$= \hat{\mu}_{t_0}(K) - 2\beta(\hat{n}_{t_0}(K)) - \sqrt{\alpha \log t}. \quad (33)$$

For the next time $t_1 \in (t_0, t]$ arm $k$ is chosen after round $t_0$, we have that the UCB of arm $k$ is higher than that of target arm $K$ in round $t_1 - 1$. However,

$$
\begin{aligned}
\hat{\mu}_{t_1-1}(k) &+ \sqrt{\frac{\alpha \log t_1}{\hat{n}_{t_1-1}(k)}} \\
&= \hat{\mu}_{t_0}(k) + \sqrt{\frac{\alpha \log t_1}{\hat{n}_{t_0}(k)}} \\
&\leqslant \hat{\mu}_{t_0}(K) - 2\beta(\hat{n}_{t_0}(K)) - \sqrt{\alpha \log t} + \sqrt{\frac{\alpha \log t_1}{\hat{n}_{t_0}(k)}} \\
&\leqslant \hat{\mu}_{t_1}(K) - \sqrt{\alpha \log t} + \sqrt{\frac{\alpha \log t_1}{\hat{n}_{t_0}(k)}} \\
&\leqslant \hat{\mu}_{t_1}(K),
\end{aligned} \tag{34}
$$

where the second inequality is due to monotonically decreasing property of $\beta$. Now, we construct the contradiction: arm $k$ does not have the highest UCB, it should not be pulled in round $t_1$. $\square$

By Lemma 3, any non-target arm $k$ is pulled for at most $0.5 \log \log t + 1 \leqslant \log \log t$ times up to round $t$. Now, if $k$ is selected in round $t$, we have Equation (32) and the following equation:

$$
\hat{\mu}_k(t) = \frac{\hat{\mu}_t^0(k)\hat{n}_t(k) - \sum_{s \in \tau(t,k)} \gamma^{(m)}(s)}{\hat{n}_t(k)}. \tag{35}
$$

Therefore, combine these two equations, we have:

$$
\begin{aligned}
\frac{1}{\hat{n}_t(k)} \sum_{s \in \tau(t,k)} \gamma^{(m)}(s) &= \hat{\mu}_t^0(k) - \hat{\mu}_t(K) + 2\beta(\hat{n}_t(K)) + \sqrt{\alpha} \exp(\hat{n}_t(k)) \\
&\leqslant \Delta(k, K) + 3\beta(\hat{n}_t(K)) + \beta(\hat{n}_t(k)) + \sqrt{\alpha} \exp(\hat{n}_t(k)) \\
&\leqslant \Delta(k, K) + 4\beta(\hat{n}_t(k)) + \sqrt{\alpha} \exp(0.5 \log \log t + 1) \\
&\leqslant \Delta(k, K) + 4\beta(\hat{n}_t(k)) + \exp \sqrt{\alpha \log t},
\end{aligned} \tag{36}
$$

where we use the monotonically decreasing property of $\beta$ and the fact of $\hat{n}_t(K) > \hat{n}_t(k)$. Thus,

$$
\begin{aligned}
\sum_{s \in \tau(T,k)} \gamma^{(m)}(s) &\leqslant \hat{n}_T(k)(\Delta(k, K) + 4\beta(\hat{n}_T(k)) + \exp \sqrt{\alpha \log T}) \\
&\leqslant (\log \log T)(\Delta(k, K) + 4\beta(\hat{n}_T(k)) + \exp \sqrt{\alpha \log T}) \\
&\leqslant (\log \log T)(\Delta(k, K) + \exp \sqrt{\alpha \log T}) + 4 \log \log T \cdot \beta(\hat{n}_T(k)) \\
&\leqslant \log \log T(\Delta(k, K) + \exp \sqrt{\alpha \log T}) + \sigma \sqrt{32 \log \log T \log \frac{\pi^2 K (\log \log T)^2}{3\delta}}.
\end{aligned} \tag{37}
$$

Finally, sum over all non-target arms, we get

$$
\begin{aligned}
\sum_{s \leqslant T} \gamma^{(m)}(s) &\leqslant \sum_{k < K} \log \log T(\Delta(k, K) + \exp \sqrt{\alpha \log T}) \\
&\quad + \sigma(K - 1) \sqrt{32 \log \log T \log \frac{\pi^2 K (\log \log T)^2}{3\delta}}.
\end{aligned} \tag{38}
$$

This ends the proof. $\square$

## B.3 ATTACKS AGAINST UCB-TCOM

As mentioned in Section 3, UCB-TCOM stands as the state-of-art algorithm in homogeneous settings. It boasts near-optimal regret, with communication costs limited to just $O(\log \log T)$. In UCB-TCOM algorithm, we assume that the optimal arm is unique, i.e., $\mu(1) > \mu(k)$ for all $k > 1$. The

---

**Algorithm 5** Attack against UCB-TCOM (Agent $m$)

---

1: **Initialization**: $\hat{\mu}_t(k) = \hat{n}_t(k) = 0$ for all $k \in [K]$
2: **for** $t = 1, 2, 3, \ldots, T$ **do**
3:      Attacker observes that agent selects $k_t^{(m)}$ by UCB
4:      Environment reveals reward $X_t^{(m,0)}(k_t^{(m)})$
5:      **if** $k_t^{(m)} \neq K$ **then**
6:          Attacker manipulates reward $X_t^{(m)}(k_t^{(m)}) = X_t^{(m,0)}(k_t^{(m)}) - \gamma^{(m)}(t)$ according to Equation (40)

---

key ideas that help UCB-TCOM algorithm to decrease the communication cost are: first, communications occur only when there are sufficient local samples for the agents; second, information about the optimal arm is not directly broadcast. In this section, we demonstrate that our attack strategy is also effective against the UCB-TCOM algorithm.

To begin with, we first discuss why we need to slightly change Equation (30). This adjustment is required due to the consideration of delayed information. Under the UCB-TCOM strategy, all $M$ agents simultaneously select an arm $k$ for multiple consecutive rounds until the number of samples $\hat{n}_t(k)$ exceeds a predefined threshold. We refer to these consecutive rounds as a *phase*. The agents share their local information and update $\hat{\mu}_t(k)$, $\hat{n}_t(k)$, and UCB values at the end of each phase. As a result, it becomes essential to compute the attack value carefully in each round, accounting for the delayed counters. Assume arm $k_t^{(m)} = k \neq K$ for all agents $m \in \mathcal{M}$, where $t$ belongs to a phase from round $s+1$ to $r$, and denote the last phase that arm $k$ is selected ends at as round $t'$. We provide the condition in Equation (30) again here:

$$\hat{\mu}_t(k) \leq \hat{\mu}_t(K) - 2\beta(\hat{n}_t(K)) - \Delta_0, \tag{39}$$

while $\hat{\mu}_t(K) = \hat{\mu}_s(K)$ and $\hat{n}_t(K) = \hat{n}_s(K)$ and they can be computed in round $t$. In addition, we can compute the true value of $\hat{\mu}_t(k)$ even if the agents will not update this value. Assume that we only attack agent $m \in \mathcal{M}$. Then, we can compute the attack values:

$$\gamma^{(m)}(t) = (\hat{\mu}_{t'}(k)\hat{n}_{t'}(k) + \sum_{h=s+1}^{t}\sum_{m'=1}^{M} X_h^{(m',0)}(k) - \sum_{h=s+1}^{t-1}\gamma^{(m)}(h)$$
$$- (\hat{n}_{t'}(k) + (t-s)M)(\hat{\mu}_t(K) - 2\beta(\hat{n}_t(K)) - \Delta_0))_+, \tag{40}$$

where $(x)_+$ represents the maximum of $x$ and $0$. It is worth noting that Equation (40) handles samples from both the previous phase and the current one separately. The latter requires special consideration due to the delayed updates. The entire process is outlined in Algorithm 5.

**Theorem 6.** *Suppose $T > K$, set the parameters of UCB-TCOM as $\beta > 1$ and $\delta < 1/2$. With probability at least $1 - \delta$, Algorithm 5 misguides the UCB-TCOM algorithm to choose the target arm $K$ at least $MT - (K-1)\left(\frac{2\beta}{\Delta_0^2}\log T\right)$ rounds, using a a cumulative attack cost at most*

$$C(T) \leqslant \left(\frac{2\beta}{\Delta_0^2}\log T\right)\sum_{k<K}(\Delta(k,K) + \Delta_0) + \frac{8(K-1)\sigma^2}{\Delta_0^2}\sqrt{\beta\log T\log\frac{4K\beta^2\pi^2(\log T)^2}{3\delta\Delta_0^4}}.$$

*Proof.* The proof is similar to Appendix A.1 and we use the same notations there. However, some steps should be modified carefully.

**Lemma 4.** *Assume event $E$ holds and $\delta \leqslant 1/2$. For any $k \neq K$ and any $t > K$, we have*

$$\hat{n}_t(k) \leqslant \min\{\beta\hat{n}_t(K), \frac{2\beta\log t}{\Delta_0^2}\} \tag{41}$$

*Proof.* Fix some $t > K$, which satisfies $k_t^{(m)} = k \neq K$ for all $m \in \mathcal{M}$, and $t$ is in a phase from round $s + 1$ to $r$. Also, denote the last phase arm $k$ is pulled is ended at round $t'$. In round $t'$, we have the following inequality by our attack design:

$$\hat{\mu}_{t'}(k) \leqslant \hat{\mu}_{t'}(K) - 2\beta(\hat{n}_{t'}(K)) - \Delta_0. \tag{42}$$

On the other hand, arm $k$ is selected in round $s + 1$ because it has the highest UCB in round $s$:

$$\hat{\mu}_s(k) + \sqrt{\frac{2 \log s}{\hat{n}_s(k)}} \geqslant \hat{\mu}_s(K) + \sqrt{\frac{2 \log s}{\hat{n}_s(K)}}. \tag{43}$$

Note that $\hat{\mu}_{t'}(k) = \hat{\mu}_s(k)$. Substituting Equation (42) into Equation (43), we have:

$$
\begin{aligned}
\sqrt{\frac{2 \log s}{\hat{n}_s(k)}} - \sqrt{\frac{2 \log s}{\hat{n}_s(K)}} &\geqslant \hat{\mu}_s(K) - \hat{\mu}_s(k) \\
&\geqslant \hat{\mu}_s(K) - (\hat{\mu}_{t'}(K) - 2\beta(\hat{n}_{t'}(K)) - \Delta_0) \\
&\geqslant \Delta_0 \\
&> 0,
\end{aligned}
\tag{44}
$$

where the third inequality is due to the monotonically decreasing property of function $\beta$. Therefore,

$$\hat{n}_t(K) = \hat{n}_s(K) \geqslant \hat{n}_s(k) = \frac{1}{\beta}\hat{n}_r(k) \geqslant \frac{1}{\beta}\hat{n}_t(k). \tag{45}$$

In addition, as the bonus term is non-negative, we have:

$$\hat{n}_t(k) \leqslant \hat{n}_r(k) = \beta\hat{n}_s(k) \leqslant \frac{2\beta \log s}{\Delta_0^2} \leqslant \frac{2\beta \log t}{\Delta_0^2}. \tag{46}$$

$\square$

**Lemma 5.** *Assume event $E$ holds and $\delta \leqslant 1/2$. For any $k \neq K$ and any $t > K$, we have*

$$\sum_{h \in \tau(t,k)} \gamma^{(m)}(h) \leqslant \hat{n}_t(k)(\Delta(k, K) + \Delta_0 + 3\beta(\hat{n}_t(K)) + \beta(\hat{n}_t(k))) \tag{47}$$

*Proof.* Note that although agents do not update their counters until each phase is over, the attacker does have the latest information thus it can maintain the latest $\hat{\mu}_t(k)$ and $\hat{n}_t(k)$ in each round $t$ even if it is not the last round of a phase. Equation (40) can be written in this form:

$$
\begin{aligned}
\gamma^{(m)}(t) = &(\hat{\mu}_{t'}^{(0)}(k)\hat{n}_{t'}(k) + \sum_{h=s+1}^{t} \sum_{m'=1}^{M} X_h^{(m',0)}(k) - \sum_{h \in \tau(t',k)} \gamma^{(m)}(h) - \sum_{h=s+1}^{t-1} \gamma^{(m)}(h) \\
&- (\hat{n}_{t'}(k) + (t - s)M)(\hat{\mu}_r(K) - 2\beta(\hat{n}_r(K)) - \Delta_0))_+,
\end{aligned}
\tag{48}
$$

where $\hat{\mu}_{t'}^{(0)}(k)$ is the global pre-attack empirical mean of arm $k$ up to round $t'$. Also, as in Appendix A.1, we only consider the round $t$ such that $\gamma^{(m)}(t) > 0$. Therefore, we have:

$$
\begin{aligned}
\sum_{h \in \tau(t,k)} \gamma^{(m)}(h) = &\hat{\mu}_{t'}^{(0)}(k)\hat{n}_{t'}(k) \\
&+ \sum_{h=s+1}^{t} \sum_{m'=1}^{M} X_h^{(m',0)}(k) - (\hat{n}_{t'}(k) + (t - s)M)(\hat{\mu}_r(K) - 2\beta(\hat{n}_r(K)) - \Delta_0) \\
= &\hat{n}_t(k)(\hat{\mu}_t^0(k) - (\hat{\mu}_t(K) - 2\beta(\hat{n}_t(K)) - \Delta_0)) \\
\leqslant &\hat{n}_t(k)(\Delta(k, K) + \Delta_0 + 3\beta(\hat{n}_t(K)) + \beta(\hat{n}_t(k))),
\end{aligned}
\tag{49}
$$

where the last inequality is due to the event $E$. $\square$

With Lemma 4, we can easily get that the target arm $K$ is selected for at least $MT - (K - 1)(\frac{2\beta}{\Delta_0^2} \log T)$ times. For the cumulative cost, we use Lemma 5, and sum over all non-target arms.

---

**Algorithm 6** Attack against DPE2 (Leader)

---

1: **Initialization**: $\hat{V}_t(k) = \hat{N}_t(k) = D_t(k) = 0$ for all $k \in [K]$, $C(t) = \emptyset$
2: **for** $t = 1, 2, 3, \ldots, T$ **do**
3:     Attacker observes that agent selects $k_t^{(1)}$
4:     Environment reveals reward $X_t^{(1,0)}(k_t^{(1)})$
5:     **if** $k_t^{(1)} \neq K$ **then**
6:         Attacker manipulates reward $X_t^{(1)}(k_t^{(1)}) = X_t^{(1,0)}(k_t^{(1)}) - \gamma(t)$ according to Equation (51)

---

Also, it is easy to get $\beta(\hat{n}_t(K)) \leqslant \beta(\frac{1}{\beta}\hat{n}_t(k))$ as $\beta()$ is a monotonically decreasing function. Therefore,

$$
\sum_{t=1}^{T} \gamma^{(m)}(t) \leqslant \sum_{k=1}^{K-1} \hat{n}_t(k)(\Delta(k, K) + \Delta_0) + 4\beta \sum_{k=1}^{K-1} \hat{n}_t(k)\beta(\frac{1}{\beta}\hat{n}_t(k))
$$

$$
\leqslant \left( \frac{2\beta}{\Delta_0^2} \log T \right) \sum_{k<K} (\Delta(k, K) + \Delta_0) + \frac{8(K-1)\beta^2 \sigma}{\Delta_0^2} \sqrt{\log T \log \frac{4K\pi^2 (\log T)^2}{3\delta\Delta_0^4}}.
$$
(50)

$\square$

### B.4    ATTACKS AGAINST LEADER-FOLLOWER ALGORITHM

In contrast to fully distributed algorithms, there exists a server, or leader (agent) in leader-follower algorithms, which has a pivotal role in exploration. On the other hand, the followers always undertake exploitation. In this section, we consider attacks on a representative leader-follower algorithm, the DPE2 algorithm, proposed by (Wang et al., 2020a). We show that our attack algorithm against fully distributed algorithms can be extended to these leader-follower algorithms.

We first introduce some new notations to differentiate between leader-follower algorithms and fully distributed algorithms. In DPE2 algorithm, the minimal mean reward gap is positive, i.e., $\Delta_{\min} = \min_{k \neq l} |\mu(k) - \mu(l)| > 0$. Without loss of generality, let agent 1 be the leader of the system. Let $\hat{N}_t(k)$ denote the times arm $k$ is selected up to time slot $t$, and $\hat{V}_t(k)$ be the post-attack empirical mean associated with $\hat{N}_t(k)$. For the ease of presentation, we consider the UCB1 induces instead of the KL-UCB induces, and define $D_t(k) := \hat{V}_t(k) + \sqrt{\frac{\alpha \log t}{2\hat{N}_t(k)}}$ as the UCB. The leader explores different arms by maintaining a list $C(t)$ which contains the suboptimal arms whose upper bounds are larger than the empirical mean of what it considers to be the optimal arm. Similar to the UCB-TCOM algorithm, the information is not updated immediately after each round. We also define the phase in the process. When $C(s - 1) = \emptyset$ and $C(s) \neq \emptyset$, we say the phase begins at round $s$; and when $C(r - 1) \neq \emptyset$ and $C(r) = \emptyset$, we say the phase ends at round $r - 1$. As the design of DPE2, the information of all arms will be updated in round $r$. We then introduce our attack algorithm.

As the followers always select the arm which the leader considers as the best, we only need to misguide the leader to regard the target arm $K$ as the optimal arm. Therefore, we only need to attack the leader. Assume $k_t^{(1)} = k \neq K$, and $t$ belongs to a phase from round $s$ to $r - 1$. Our attacks make sure:

$$
\hat{V}_r(k) \leqslant \hat{V}_s(K) - 2\beta(\hat{N}_s(K)) - \Delta_0,
$$
(51)

where $\hat{V}_r(k) = \frac{\hat{V}_s(k)\hat{N}_s(k) + X_t^{(1,0)}(k) - \gamma(t)}{\hat{N}_r(k)}$, and the attack value is $\gamma(t)$. The details are described in Algorithm 6.

**Theorem 7.** *Suppose $T > T_0$, and $\delta < 1/2$. With probability at least $1 - \delta$, Algorithm 6 misguides the DPE2 algorithm to choose the target arm $K$ at least $M(T - K) - (K - 1)\left( \frac{\alpha}{2\Delta_0^2} \log T + 1 \right)$*

*rounds, using a a cumulative attack cost at most*

$$C(T) \leqslant \left( \frac{\alpha}{2\Delta_0^2} \log T + 1 \right) \sum_{k < K} (\Delta(k, K) + \Delta_0)$$

$$+ 4(K-1)\sigma \sqrt{2(\frac{\alpha}{2\Delta_0^2} \log T + 1) \log(\frac{K\pi^2}{3\delta}(\frac{\alpha}{2\Delta_0^2} \log T + 1)^2)},$$

*where $T_0 / \log(T_0) = K \lceil \frac{\alpha}{2\Delta_0^2} + 1 \rceil$.*

*Proof.* The proof is similar to Appendix A.1 as well.

**Lemma 6.** *Assume event $E$ holds and $\delta \leqslant 1/2$. For any $k \neq K$ and any $t > T_0$, we have*

$$\hat{N}_t(k) \leqslant \frac{\alpha}{2\Delta_0^2} \log t + 1$$

*Proof.* Fix some $t > T_0 \geqslant K$, which satisfies $k_t^{(1)} = k \neq K$, and $t$ is in a phase from round $s$ to $r-1$. Also, assume the last phase arm $k$ is pulled from round $s'$ to $r'-1$. In round $r'$, we have the following inequality by the design of our attacks:

$$\hat{V}_{r'}(k) \leqslant \hat{V}_{s'}(K) - 2\beta(\hat{N}_{s'}(K)) - \Delta_0. \tag{52}$$

On the other hand, arm $k$ is selected in round $s$ because the following inequality holds in round $s$:

$$\hat{V}_s(k) + \sqrt{\frac{\alpha \log s}{\hat{N}_s(k)}} \geqslant \hat{V}_s(K). \tag{53}$$

Note that $\hat{V}_s(k) = \hat{V}_{r'}(k)$. Substituting Equation (52) into Equation (53), we have:

$$\sqrt{\frac{\alpha \log s}{\hat{N}_s(k)}} \geqslant \hat{V}_s(K) - \hat{V}_s(k)$$

$$\geqslant \hat{V}_s(K) - (\hat{V}_{s'}(K) - 2\beta(\hat{N}_{s'}(K)) - \Delta_0) \tag{54}$$

$$\geqslant \Delta_0,$$

where the third inequality is due to the monotonically decreasing property of $\beta$. Therefore, this ends the proof as $\hat{N}_t(k) \leqslant \hat{N}_s(k) + 1$. $\square$

**Lemma 7.** *Assume event $E$ holds and $\delta \leqslant 1/2$. For any $k \neq K$ and any $t > K$, we have*

$$\sum_{h \in \tau(t,k)} \gamma(h) \leqslant \hat{N}_t(k)(\Delta(k, K) + \Delta_0 + 3\beta(\hat{N}_t(K)) + \beta(\hat{N}_t(k))) \tag{55}$$

*Proof.* Similar to Equation (48), the attack value can be written in this form:

$$\gamma(t) = \left( \hat{V}_s^{(0)}(k)\hat{N}_s(k) + X_t^{(1,0)}(k) - \sum_{h \in \tau(s,k)} \gamma(h) - (\hat{N}_s(k) + 1)(\hat{V}_s(K) - 2\beta(\hat{N}_s(K)) - \Delta_0) \right)_+, \tag{56}$$

where $\hat{V}_s^{(0)}(k)$ is the global pre-attack empirical mean of arm $k$ up to round $s$ (for the leader). Also, as in Appendix A.1, we only consider the round $t$ such that $\gamma(t) > 0$. Therefore, we have:

$$\sum_{h \in \tau(t,k)} \gamma(h) = \hat{V}_s^{(0)}(k)\hat{N}_s(k) + X_t^{(1,0)}(k) - (\hat{N}_s(k) + 1)(\hat{V}_s(K) - 2\beta(\hat{N}_s(K)) - \Delta_0)$$

$$= \hat{N}_t(k)(\hat{V}_t^0(k) - (\hat{V}_t(K) - 2\beta(\hat{N}_t(K)) - \Delta_0)) \tag{57}$$

$$\leqslant \hat{N}_t(k)(\Delta(k, K) + \Delta_0 + 3\beta(\hat{N}_t(K)) + \beta(\hat{N}_t(k))),$$

where the last inequality is due to the event $E$. $\square$

With Lemma 6, we can easily get that the target arm $K$ is selected for at least $M(T - K) - (K - 1)\left(\frac{\alpha}{2\Delta_0^2}\log T + 1\right)$ times because in the beginning, followers randomly select an arm to pull, and after that, $K$ will be the optimal arm for the leader after each phase, so that followers won't select arms other than $K$. As for the cumulative cost, we introduce Lemma 7, and sum over all non-target arms. Also, it is easy to get $\beta(\hat{N}_t(K)) \leqslant \beta(\hat{N}_t(k))$ as $\beta$ is a monotonically decreasing function and $T > T_0$, which means $\hat{N}_t(k) \leqslant \hat{N}_t(K)$ holds for any $k \neq K$. Therefore,

$$
\begin{aligned}
\sum_{t=1}^{T} \gamma(t) &\leqslant \sum_{k=1}^{K-1} \hat{N}_t(k)(\Delta(k, K) + \Delta_0) + 4\beta \sum_{k=1}^{K-1} \hat{N}_t(k)\beta(\hat{N}_t(k)) \\
&\leqslant \left(\frac{\alpha}{2\Delta_0^2}\log T + 1\right) \sum_{k < K} (\Delta(k, K) + \Delta_0) \\
&\quad + 4(K-1)\sigma\sqrt{2(\frac{\alpha}{2\Delta_0^2}\log T + 1)\log(\frac{K\pi^2}{3\delta}(\frac{\alpha}{2\Delta_0^2}\log T + 1)^2)}.
\end{aligned}
\tag{58}
$$

$\square$

## B.5 ATTACKS AGAINST GENERAL ALGORITHMS

In this section, we provide a more detailed explanation of the general attack strategy. We consider general CMA2B algorithms satisfying the following assumptions.

**Assumption 1.** *Each agent in the CMA2B algorithm immediately shares its reward observations with the others, ensuring that all agents maintain identical empirical means for making decisions.*

**Assumption 2.** *The CMA2B algorithm chooses suboptimal arms no more than $R(T) = o(T)$ times for $T$ rounds,*

Note that, for ease of presentation, we disregard the effects of delayed communications using Assumption 1. However, techniques provided in Appendix B.3 can be applied to address scenarios with limited communication.

We then introduce the general attack strategy. As in the previous sections, we can arbitrarily select an agent $m$ as our target agent. In round $t$, if the chosen agent $m$ plays arm $k$ that is not the target arm $K$, we manipulate its reward to be

$$
X_t^{(m)}(k) = X_t^{(m,0)}(k) - \gamma_t^{(m)},
\tag{59}
$$

where the attack value $\gamma_t^{(m)}$ is calculated by

$$
\gamma_t^{(m)} = [\hat{\mu}_{t-1}(k) - \hat{\mu}_{t-1}(K) + \beta(\hat{n}_{t-1}(k)) + \beta(\hat{n}_{t-1}(K))]_+ .
\tag{60}
$$

Based on this attack design, we provide the following theorem.

**Theorem 8.** *Consider an arbitrary no-regret CMA2B algorithm satisfying Assumptions 1 and 2. With probability at least $1 - \delta$, the general attack strategy will mislead the CMA2B algorithm to choose the target arm $K$ at least $T - R(T)$ rounds. Its cumulative attack cost is bounded by*

$$
C(T) = \sum_t |\gamma_t^{(m)}| \leqslant O\left(\sum_{k \neq K}(\Delta(k, K) + 4\beta(1))R(T)\right).
\tag{61}
$$

Compared to the result in Theorem 1, the $\beta()$ term in this cumulative cost is $\beta(1)$, which can be significantly larger. This suggests that the general attack strategy incurs a higher cost compared to attack strategies tailored specifically for UCB-based algorithms.

*Proof.* We again define the "good event" $E = \{\forall k, \forall t > K : |\hat{\mu}_t(k) - \mu(k)| < \beta(\hat{n}_t(k))\}$, where $\hat{\mu}_t(k)$ is the pre-attack empirical mean of arm $k$ for all agents up to time slot $t$. By Hoeffding's inequality, we can prove that for any $\delta \in (0, 1)$, $P(E) > 1 - \delta$. With event $E$, we have that for any arm $k \neq K$,

$$
[\mu(k) - \mu(K)]_+ < [\hat{\mu}_t(k) - \hat{\mu}_t(K) + \beta(\hat{n}_t(k)) + \beta(\hat{n}_t(K))]_+.
\tag{62}
$$

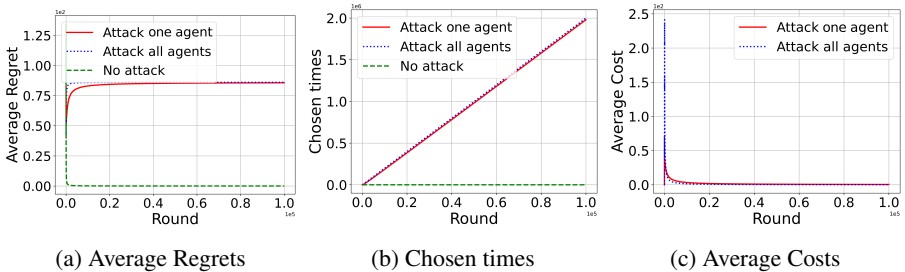

(a) Average Regrets       (b) Chosen times       (c) Average Costs

Figure 4: Attacks against UCB.

Since the attack value $\gamma_t^{(m)}$ equals to the right hand side, the best arm is now the target arm $K$. With Assumption 2, the expected number of pulling the target arm is

$$\mathbb{E}[\hat{n}_T(K)] = T - R(T). \tag{63}$$

With event $E$, we also have

$$\hat{\mu}_t(k) - \hat{\mu}_t(K) < \mu(k) - \mu(K) + \beta(\hat{n}_t(k)) + \beta(\hat{n}_t(K)), \tag{64}$$

which implies

$$\gamma_t^{(m)} < [\mu(k) - \mu(K) + 2\beta(\hat{n}_t(k)) + 2\beta(\hat{n}_t(K))]_+ \leqslant \Delta(k, K) + 2\beta(\hat{n}_t(k)) + 2\beta(\hat{n}_t(K)). \tag{65}$$

Since $\beta()$ is a decreasing function, we have

$$\sum_t |\gamma_t^{(m)}| = \sum_{k \neq K} (\Delta(k, K) + 4\beta(1))\hat{n}_T(k) \leqslant O\left(\sum_{k \neq K} (\Delta(k, K) + 4\beta(1))R(T)\right). \tag{66}$$

$\square$

### B.6 Additional Experiments

In this section, we show the experimental results of attacks against the CO-UCB algorithm in homogeneous settings. We set $T = 100,000, K = 20, M = 20$. The distributions of arms are the same as those in Section 5. CO-UCB takes $\alpha = 4$, and attack parameters are set to $\Delta_0 = 0.1$, and $\delta = 0.1$. We compare our algorithm, which only attacks one agent, with two baselines: the first one attacks all agents using attack values computed by Equation (30) for each agent; the second one is the original CO-UCB algorithm without attacks. Each experiment was repeated for 10 times.

In Figures Figure 4a and Figure 4b, the CO-UCB algorithm displays sublinear regret; the curve showcasing its regret gradually approaches 0, and the algorithm seldom opts for the suboptimal target arm $K$. However, both attack strategies successfully misguide the CO-UCB algorithm, leading it to consistently select our target arm, as shown in Figure 4b. This causes linear regrets, as highlighted in Figure 4a. Notably, the divergence between these attack algorithms in both figures is minimal, suggesting that even though our attack is designed to target just one agent, it is almost as effective as a strategy targeting all agents. Looking at Figure 4c, the cumulative attack costs for both strategies are nearly indistinguishable, converging towards 0. This points to sublinear costs for both. Such a finding amplifies the effectiveness of our attack design: despite focusing on a single agent, the cumulative cost is nearly identical to an approach targeting all 20 agents.

