# OpenReview forum: "Adversarial Attacks on Cooperative Multi-agent Bandits"
_ICLR.cc/2025/Conference — Submitted to ICLR 2025_

### Official Review · Reviewer_JBeq · 2024-10-30

**Soundness:** 1
**Presentation:** 1
**Contribution:** 2
**Rating:** 3
**Confidence:** 4

**Summary:**

This paper presents attack strategies for collaborative multi-armed bandits using CO-UCB algorithms under two distinct settings: a homogeneous setting, where all agents have access to all arms, and a heterogeneous setting, where different agents have access to different subsets of arms. For the heterogeneous setting, the author considers two types of attackers: an oracle attacker, who has explicit knowledge of the average reward ranking of each arm, and a non-oracle attacker, who lacks this information. Additionally, the author conducts experiments to validate the theoretical results.

**Strengths:**

This paper is the first to explore adversarial attacks in the context of cooperative learning within multi-agent, multi-armed bandits. This problem is novel. Furthermore, the design of the algorithms for Affected Agents Selection (AAS), Target Agents Selection (TAS), and the proof of Theorem 3 are both innovative and nontrivial.

**Weaknesses:**

First, the paper lacks discussion about the novelty and major technical challenges for various results (e.g., Theorem 1, Proposition 1 and 2), which makes it really difficult to judge the significance of the contribution. For instance, they state: "In Section 3, for homogeneous CMA2B, we devise attack strategies for three representative algorithms by targeting a single agent to misguide all agents. We prove that the attack costs are independent of the number of agents, M." However, since observations are shared and all agents are homogeneous, it is unsurprising that attacking one agent could misguide the rest, making the cost independent of M. This result appears intuitively straightforward. Additionally, the attack strategy seems quite similar to that in Zuo, S. (2024, April), with the proof being a straightforward extension from the single-agent to the multi-agent setting. The authors do not highlight any significant additional challenges, making it unclear how significant the technical contributions are.

Second, many results and proofs are inaccurate, and require the reader’s careful verification in order to convince themselves that the proof or statement is really correct. Even I was able to verify that some missing proofs can be fixed, I was not able to check all the proofs and it makes me feel the paper really needs significant amount of re-writing and polishing before being ready for publication.
1.	In the statement of Theorem 1, the term "our attack strategy" is ambiguous because there is only some vague descriptions about the attack idea and no clear mathematical definition about the attack. The reader had to go to the appendix to check the definition.
2.	In the proof of Theorem 2 (page 16), it states “\(f\) is submodular and the greedy algorithm in Algorithm 1 gives a \((1 - 1/e)\)-approximation solution”. This is inaccurate – the paper needs to additional prove the function is monotone and non-negative (PS: the paper also need to cite papers for this statement as not everyone knows this result). I think I convinced myself that these two properties are also true, but I think a maturely written paper should not ask reviewers to complete these reasonings and verify correctness.
3.	In Theorem 3 (also Theorem 4), the term \(\Delta(k, k+1)\) is not defined rigorously, as the author has not established that \(K \notin K_0\). If \(K \in K_0\), \(\Delta(K, K+1)\) is undefined. Also, Equation (3) does not include \(T_0\); actually none of the equations in this page has T_0. In the statement of Theorem 3, the definition of \(g(k)\) is missing, which also affects the subsequent discussion. I had to go to the appendix to double check the correct expression and definition


In addition to above concerns on the significance of contributions, the paper contains a few other major issues, including too many syntax issues, gaps in proofs, too much assumptions of prior knowledge from the reader, and a lack of discussion on the contributions (including over-claiming contributions). These factors make it difficult for readers to fully understand the key ideas and may lead them to question the rigor of the proofs. A few examples are listed as follows.

1.	Line 111 to 112, the paper claims that “Our work is the first to study how an attacker may manipulate multi-agent cooperative learning”. However, this ICML’20 https://proceedings.mlr.press/v125/boursier20a.html, Mobihoc’21 paper ahttps://dl.acm.org/doi/abs/10.1145/3466772.3467045 and AAMAS’22 https://dl.acm.org/doi/10.5555/3635637.3662992 seem to have already been studying similar problem of attacking cooperative bandit problem.

2.	In line 6 of Algorithm 1, what is \(\omega(k)\)

3.	On Page 5, line 268, the stated goal is to “leverage sublinear attack costs to misguide the maximum number of agents, aiming to increase the overall count of agents enduring linear regrets.” However, why using Algorithm 3 will lead to “attacking the maximum number of agents” is neither justified nor adequately explained in the subsequent paragraphs, leaving readers to infer the reasoning on their own.

4.	The second claimed contribution appears to be inaccurate. The paper states, "linear costs are necessary to fool all agents into suffering linear regrets." However, this seems straightforward, as the target arm attack might not be a suitable objective in a heterogeneous environment (as mentioned in line 252). Therefore, a more accurate contribution might be to propose an appropriate and meaningful "attacking goal" specifically tailored to the heterogeneous setting, based on my understanding.

5.	On page 7, line 327, the phrase “is it feasible to attack this group using only a limited number of agents” is unclear. It would be beneficial to clarify what is meant by “using a limited number of agents.” If the intended meaning is “attacking a limited number of agents to misguide the maximum number of agents without conflicts, resulting in linear regret,” then it should be explicitly stated to avoid confusion.

6.	On page 9, line 463, the term \( \text{UCB}_t(k) \) is introduced without being defined (though the reviewer knows, but this is not a proper way to write a general-purpose ML paper), and there is no reference to this quantity elsewhere in the text.

7.	Figure 3 lacks standard deviations or confidence intervals, which undermines the claim that “Both LTA and OA result in the most significant average regrets.” Without measures of variability, it is difficult to assess the statistical reliability and significance of this claim. To strengthen the argument, it would be beneficial to include standard deviations or confidence intervals in the figure, providing a clearer depiction of the uncertainty and supporting the robustness of the results.

**Questions:**

Regarding the problem setting, what is the difference between this paper’s attack setting and previous works such as ICML’20 https://proceedings.mlr.press/v125/boursier20a.html, Mobihoc’21 paper ahttps://dl.acm.org/doi/abs/10.1145/3466772.3467045 and AAMAS’22 https://dl.acm.org/doi/10.5555/3635637.3662992.(and possibly others since I only did a light search about “attacking multi-agent bandits” on Google Scholar)?

It seems to the reviewer that the homogeneous agent setting is a simple corollary of Zuo’24 result. Would it be possible to just describe Theorem 1 as a simple corollary of Zuo’24? What’s the research value of theorem 1?

Can we eliminate TAS? What is the benefit of including TAS? Currently Theorem 3 shows that the attack cost is independent of the number of affected agents by attacking limited number of agents generated by TAS, what if we just attack agents in K0? What will change? Will it significantly inflate the attacking cost?

Are there any real-world applications, aside from “botnets,” that can be modeled as adversarial attacks on cooperative multi-agent bandits? The introduction currently provides only one example and lacks supporting references.

---

> ### Author Response · Authors · 2024-12-02
>
> **Q1.** *First, the paper lacks discussion about the novelty and major technical challenges for various results.*
>
> **A1.** Regarding the homogeneous setting, please refer to A1 to Reviewer bPAF. For the heterogeneous setting, our work tackles several novel challenges in heterogeneous cooperative learning. These include identifying meaningful attack objectives and developing the AAS and TAS algorithms specifically tailored to heterogeneous environments. Furthermore, the theoretical analysis of our attack strategies is particularly challenging due to a unique aspect of the multi-agent heterogeneous setting: target agents may not consistently have sufficient attack opportunities. Resolving this issue requires a careful treatment, as detailed in Lines 383–404.
>
> **Q2.** *Second, many results and proofs are inaccurate. 1. In the statement of Theorem 1, the term "our attack strategy" is ambiguous. 2. “(f) is submodular and the greedy algorithm in Algorithm 1 gives a ((1 - 1/e))-approximation solution”. This is inaccurate – the paper needs to additional prove the function is monotone and non-negative. 3. In Theorem 3 (also Theorem 4), the term $\Delta(k, k+1)$ is not defined rigorously. Also, Equation (3) does not include $T_0$. The definition of $g(k)$ is missing.*
>
> **A2.**
> 1. Due to the space limit and since we treat the homogeneous setting as the warm-up result, we defer the detailed algorithm and analysis to Appendix B.1.
>
> 2. Thank you for pointing this out. $f$ is indeed both monotone and non-negative, as a larger input set of agents will always result in more conflict-free agents. We will revise the proof to explicitly demonstrate these properties and add citations in the updated version.
>
> 3. The definition of $\Delta(k, k') = \mu(k) - \mu(k')$, which represents the difference in expected rewards between any two arms $k$ and $k'$, is provided in Line 205. $T_0$ is a constant defined as the feasible solution to Eq. (3) with respect to $t$, where the right-hand side is a problem-specific coefficient independent of $t$. As for $g(k)$, it denotes the target agent assigned to attack arm $k$, as determined in Algorithm 2 (Line 7).
>
> **Q3.** *Line 111 to 112, the paper claims that “Our work is the first to study how an attacker may manipulate multi-agent cooperative learning”. Regarding the problem setting, what is the difference between this paper’s attack setting and previous works such as ICML’20, Mobihoc’21 and AAMAS’22?*
>
> **A3.** Our intention was to claim that we are the first to explore the vulnerabilities of cooperative multi-agent bandits specifically from the attacker's perspective, extending the single-agent attack model studied in the adversarial attack literature (e.g., Jun et al., 2018; Liu \& Shroff, 2019; Zuo, 2024).
> We acknowledge that related works, such as ICML’20 and Mobihoc’21, examine settings involving honest and malicious (or selfish) agents in cooperative bandit frameworks. However, these studies do not approach the problem from an attacker's perspective aiming to mislead the entire learning system.  While AAMAS’22 investigates how attackers can exploit "collisions" to influence Multi-Player Multi-Armed Bandits (MMAB), our work is distinct in that it follows the Cooperative Multi-Agent Multi-Armed Bandit (CMA2B) framework (Vial et al., 2021; Yang et al., 2021, 2022; Wang et al., 2022, 2023a,b), which assumes no collisions and emphasizes cooperative interactions.
>
> **Q4.** *In line 6 of Algorithm 1, what is $\omega(k)$?*
>
> **A4.** $\omega(k)$ is the sorted index corresponding to the $k$-th largest set of agents.
>
> **Q5.** *On Page 5, line 268, the stated goal is to “leverage sublinear attack costs to misguide the maximum number of agents, aiming to increase the overall count of agents enduring linear regrets.” Why using Algorithm 3 will lead to “attacking the maximum number of agents” is neither justified nor adequately explained in the subsequent paragraphs.*
>
> **A5.** Sorry for the confusion. The stated goal is achieved since the AAS algorithm identifies an approximately maximum set of agents that can be attacked without conflicts. By leveraging this set, Algorithm 3 focuses on attacking the largest possible group of agents while maintaining a sublinear attack cost.
>
> **Q6.** *The second claimed contribution appears to be inaccurate.*
>
> **A6.** We appreciate your suggestion regarding the second claimed contribution, and we will revise it to emphasize our identification of an appropriate and meaningful attack objective specifically tailored to the heterogeneous setting. However, we also want to highlight that our findings and the accompanying proofs, demonstrating the necessity of linear costs to achieve either the target arm attack or linear regrets for all agents, are non-trivial and represent novel contributions to the study of adversarial attacks on CMA2B.

---

> > ### Author Response · Authors · 2024-12-02
> >
> > **Q7.** *On page 7, line 327, the phrase “is it feasible to attack this group using only a limited number of agents” is unclear.*
> >
> > **A7.** Thank you for your suggestion. We agree that your phrasing, “attacking a limited number of agents to misguide the maximum number of agents without conflicts, resulting in linear regret” is more accurate, and we will incorporate this updated phrasing in the revised version.
> >
> > **Q8.** *On page 9, line 463, the term $\text{UCB}_t(k)$ is introduced without being defined.*
> >
> > **A8.** Thank you for pointing this out. We will ensure the UCB term is explicitly defined.
> >
> > **Q9.** Figure 3 lacks standard deviations or confidence intervals.
> >
> > **A9.** Thank you for your valuable feedback regarding the experimental results. We agree that including measures of variability is essential and will include error bars in the updated version.
> >
> > **Q10.**  It seems to the reviewer that the homogeneous agent setting is a simple corollary of Zuo’24 result. Would it be possible to just describe Theorem 1 as a simple corollary of Zuo’24? What’s the research value of theorem 1?
> >
> > **A10.** Please see A1 to Reviewer bPAF.
> >
> > **Q11.** *Can we eliminate TAS? What is the benefit of including TAS? Currently Theorem 3 shows that the attack cost is independent of the number of affected agents by attacking limited number of agents generated by TAS, what if we just attack agents in $\mathcal{K}_0$? What will change? Will it significantly inflate the attacking cost?*
> >
> > **A11.** Thank you for the insightful question. TAS is essential because it selects one target agent for each local optimal arm, identifying the minimal set of target agents necessary to achieve the attack objectives in the worst-case setting (see A5 to Reviewer svoh for further discussion).
> >
> > Regarding the question about attacking $\mathcal{D}_0$ directly: $\mathcal{K}_0$ represents the set of local optimal arms, and we assume the reviewer is referring to attacking all agents in $\mathcal{D}_0$, the group of agents associated with $\mathcal{K}_0$. If we were allowed to attack all agents in $\mathcal{D}_0$, the attack cost would still be $O(|\mathcal{K}_0| \log T)$. However, this approach would require taking control of a significantly larger number of agents within the system, which may not be practical or feasible in many scenarios. By leveraging TAS, the attack is optimized to target the minimal number of agents while achieving the same effect, making the strategy more efficient in practice.
> >
> > **Q12.** *Are there any real-world applications, aside from “botnets,” that can be modeled as adversarial attacks on cooperative multi-agent bandits?*
> >
> > **A12.** Please see A1 to Reviewer n2tK.

---

### Official Review · Reviewer_n2tK · 2024-11-02

**Soundness:** 3
**Presentation:** 3
**Contribution:** 2
**Rating:** 5
**Confidence:** 3

**Summary:**

This paper studies adversarial attacks in a multi-armed bandit setting, where multiple learners are collaboratively learning. The agents may communicate their shared information between themselves. An adversary may, however, sit between the environment generating rewards and the agent and specify erroneous rewards to them. The adversary wishes to derail the agents while minimizing the number of times they need to intervene. The paper studies two models - *homogeneous*, where any agent can pull any arm and *heterogeneous* where an agent may pull only a subset of arms (determined exogenously). The first setting is easy for the adversary and using similar ideas from literature, the paper gives a nearly optimal bound. The second setting is harder, and the paper gives a set of results under different adversary goals.

In general, I found the work well-written and clearly explained and exposited. Taking the model as a given, the paper studies the reasonable set of questions that arise. That said, I am not convinced of the underlying model of collaboration or adversary that is presented; while I am quite familiar with bandit literature, this sub-category of bandit literature is not as known to me and I am unsure about how to correctly contextualize the results and reason about the major takeaways. As such, I am borderline on the work.

**Strengths:**

The work is well-written and quite easy to follow and quickly gain the main insights. The authors do a good job of highlighting their assumptions and the model, and then investigating a lot of the natural questions that emerge. If we take the model as a given, the contributions are sound.

I found the impossibility results in the heterogenous setting particularly insightful. It almost suggests that for a goal-oriented adversary (and not a purely malicious one) it is quite difficult to achieve their target without incurring high cost.

**Weaknesses:**

My main concern is the applicability or relevance of the model. What exactly is the setting where one would encounter the scenario proposed and thus this work would be insightful? Could you provide some motivating examples? Specifically, the heterogenous setting (the main meat of the paper) where an oracle attacker is assumed. The setting here afaik is as follows:
* A number of agents are collaboratively learning the same online learning with unlimited communication, but a powerful adversary, who knows the true ranking of all arms, can inject arbitrary errors into the observations of any agent?

In the heterogenous section, you essentially argue for an attacker's goal as misleading agents, and not necessarily get their (attacker's) sub-opt arm picked. In this sort of purely adversarial, almost zero-sum type of attacker reasonable? Under what context could we expect such as attacker? Or do we gather any technical/conceptual insights under this model?

In the paper, it is assumed that the attacker knows the algo used by the learner. While it can be relaxed in the homogenous case, it is necessary in the heterogenous case. I am happy that the authors explicitly mention this, but given that the heterogenous case is the main contribution of the paper, I still find this quite a big assumption and somewhat unreasonable. Are there settings you can motivate where this might be the case?

**Questions:**

See the above weaknesses.

It is likely that the agents know the distribution type for the rewards (even if they don't know the expected value or anything else). In that setting, the attacker injecting strategically chosen errors (without any real constraint) might tip the agent to think something is amiss since the samples may not reflect the distribution family they know the problem to have. Do you think it's possible - specifically in the attacks you have discussed? In general, there might be reasonable constraints to add on what can be injected.

---

> ### Author Response · Authors · 2024-12-02
>
> **Q1.** *My main concern is the applicability or relevance of the model. What exactly is the setting where one would encounter the scenario proposed and thus this work would be insightful? Could you provide some motivating examples?*
>
> **A1.** Thank you for your thoughtful question. To address the applicability and relevance of our model, we provide a motivating example in the context of online advertisements on e-commerce platforms (e.g., Amazon, eBay). In this scenario, recommendation systems on different platforms can be viewed as learning agents, and the arms represent products recommended to users. These learning systems, often belonging to the same advertising company, share user feedback with each other to collaboratively improve their understanding of user preferences. This naturally creates a cooperative learning problem.
>
> In this context, the two attack objectives studied in this paper can be interpreted as follows.
>
> - Target Arm Attack Objective (e.g., in the homogeneous setting): One product (or its associated seller) might aim to manipulate the recommender systems to increase its exposure to users. This could be achieved by targeting a platform (agent) and injecting fake user feedback (commonly known as click fraud). Our theoretical findings suggest that it is feasible for an attacker to influence the entire system with a small attack cost by leveraging such strategies. This highlights the potential risks of subtle manipulation in cooperative learning environments.
>
> - Linear Regret Objective (e.g., in the heterogeneous setting):
> In this scenario, the attacker could be a competing advertising company aiming to undermine the reputation and performance of the recommendation system. The competitor may have already collected user preference data through its own systems and could potentially infer the original company’s recommendation algorithm using simulated or fake users. Based on our findings, the competitor can strategically target a limited number of platforms to mislead the original company’s learning system, again at a low cost. Notably, our algorithms remain effective even when the set of attackable agents is restricted (footnote 1 on page 3), so the competitor does not need to access all platforms.
>
> These attack objectives are not limited to online advertisements but extend to other domains where competition exists either between arms (e.g., products, services) or between learning systems (e.g., service providers). For instance, in cooperative navigation systems where autonomous vehicles share data to optimize routes, a malicious actor could manipulate shared information to disrupt traffic flow or bias route selection toward specific paths. Similarly, in sensor networks used for cooperative source localization or environmental monitoring, an attacker could interfere with a subset of sensors to mislead the system and obscure the true source of the signal or data.
>
> In summary, the scenarios explored in our paper are relevant to a variety of cooperative learning systems. The insights gained from studying adversarial attacks in this framework contribute to a deeper understanding of vulnerabilities and the development of robust systems. We will expand on these examples and motivations in the revised version to better contextualize our contributions.
>
> **Q2.** *In the paper, it is assumed that the attacker knows the algo used by the learner.*
>
> **A2.** Please see A5 to Reviewer jeLP.

---

> > ### Author Response · Authors · 2024-12-02
> >
> > **Q3.** *It is likely that the agents know the distribution type for the rewards (even if they don't know the expected value or anything else). In that setting, the attacker injecting strategically chosen errors (without any real constraint) might tip the agent to think something is amiss since the samples may not reflect the distribution family they know the problem to have. Do you think it's possible - specifically in the attacks you have discussed? In general, there might be reasonable constraints to add on what can be injected.*
> >
> > **A3.** Thank you for the insightful observation. We acknowledge that in certain scenarios, agents with prior knowledge of the reward distribution type might detect discrepancies introduced by injected errors. However, we address this concern as follows:
> >
> > (a) In the existing body of single-agent attack studies (e.g., Jun et al., 2018; Liu \& Shroff, 2019; Zuo, 2024), such constraints are typically not considered, as these works primarily focus on unbounded continuous reward cases. Our study extends this framework to the multi-agent setting, where our attack strategy subtly shifts reward values, exploiting the inherent noise in reward observations. This makes it unlikely for agents to detect the attack, as the changes blend into the natural variability of the reward distributions.
> >
> > (b) For discrete or binary reward settings, we recognize that these constraints may require adjustments to the attack strategy. For instance, in binary reward scenarios, techniques similar to those proposed by Zuo et al. (2024) could be adapted to craft attack values that align with the known distribution family while maintaining the attack's effectiveness.

---

### Official Review · Reviewer_svoh · 2024-11-03

**Soundness:** 3
**Presentation:** 1
**Contribution:** 2
**Rating:** 5
**Confidence:** 3

**Summary:**

This work examines vulnerabilities in cooperative multi-agent multi-armed bandit systems, where agents work together to maximize rewards but can be disrupted by adversarial attacks targeting a few agents’ feedback. In homogeneous settings, where agents share the same choices, the authors show that minimal manipulation of one agent’s observations can influence the entire group to select suboptimal arms. In heterogeneous settings, with agents having different options, the authors propose strategies to induce linear regret among the maximum number of agents by attacking a few key agents. In addition to theoretical results, the authors support their claims with experimental evaluations.

**Strengths:**

- **Originality:** While previous works have dealt with robustness in multi-agent bandit systems against weak corruptions [1,2], this paper addresses weaknesses against adversarial attacks tailored for this setting. The attack target for the heterogeneous setting is novel compared to single-agent bandits [3].
- **Quality:** The theoretical analysis is rigorous, with clear proofs supporting the various claims and an experimental evaluation section.
- **Clarity:** Using simple example cases, the paper explains the motivation and intuition behind using a different attack target than in previous works.
- **Significance:** By highlighting the potential for low-cost adversarial disruptions in cooperative learning systems, the paper reveals critical vulnerabilities that could have implications for designing more robust systems.

[1] Vial, Daniel, Sanjay Shakkottai, and R. Srikant. "Robust multi-agent multi-armed bandits." Proceedings of the Twenty-second International Symposium on Theory, Algorithmic Foundations, and Protocol Design for Mobile Networks and Mobile Computing. 2021.

[2] Dubey, Abhi, and Alex Pentland. "Private and Byzantine-Proof Cooperative Decision-Making, 20th Conf. Autonomous Agents and Multiagent Systems, 2020.", 2020.

[3] Jun, Kwang-Sung, et al. "Adversarial attacks on stochastic bandits." Advances in neural information processing systems 31 (2018).

**Weaknesses:**

**Minor issues:**
- line 47: hope->hopes
- line 68: since this point is addressed in this work, the phrase 'remain uncertain' is confusing.
- line 75: 'remains an unresolved issue' - similar to my last remark.
- line 101: 'with its cost analysis'-> 'and analyze its cost'.
- line 131: the use of 'observes' is confusing here, as this is the pre-attack value and is not necessarily observed as is.
- line 139: 'collision'-> collision model.
- lines 143-152: The authors should explain this choice for the heterogeneous model, or cite previous works that use it.
- lines 191,423: 'no-regret'-> sublinear regret
- section 4.1: the intuitive examples could be explained better - how many arms does each agent have?
- lines 352-361: the term 'attack value' was never formally defined, nor did the mean $\hat{\mu}$.
- line 362: 'constant fulfills'-> constant that fulfills.
- line 424: 'affected'-> target.
- line 449: 'unknown'->known.
- line 484: 'when'-> at
- line 519: 'Leaning'-> Learning

**Other issues:**
- This work is not written clearly and assumes the reader possesses extensive technical knowledge about adversarial attacks in bandits. For example, the term attack value is not explained in the main text, and the idea behind equation (2) is also unclear. The parameters in Thm 1 are undefined as well: What are $\alpha,\Delta_0$?
- Table 1 is misleading. For the first and second lines, it is unclear whether knowledge about the time horizon $T$ is needed. The authors can cite [1] rather than [2] if it is not. More importantly, the third line assumes oracle knowledge of the attacker about arm rankings (Thm 3), which is a strong assumption. When using the version that does not use this assumption, the attack cost becomes much larger at $O(K\log T/\Delta_{\min}^2)$ (Thm 4).
- The problem addressed in this paper could be motivated better. In the Introduction, the authors mention advertisers that share information about users as an example of a multi-agent bandit system that can be attacked. However, I am not sure how practical this problem is given the requirements for user privacy preservation. It would be nice if the authors could expand on that, provide references, or suggest additional motivation.
- While the attack design for heterogeneous systems is novel, it should be mentioned that the attack itself used in this paper
technically resembles techniques from previous works [1,2] (for instance, equation (2)).
- The difficulty of adversarial attacks on a heterogeneous multi-agent system is not fully characterized. The authors mention it is not possible to cause all agents to suffer linear regrets, and find a nearly maximal set of affected agents. Regarding the target agents group, it is unclear to me if the one used in this paper is the minimal one, or if we can use fewer target agents with the same effect. In addition, as mentioned above, the attack cost under realistic assumptions is $O(K\log T/\Delta_{\min}^2)$ for the heterogeneous case, but there is no lower bound to show how good this result is.
- In the heterogeneous case, an attack is only designed for one algorithm. For single-agent bandits, it is acceptable to design attacks for very basic algorithms like UCB and $\epsilon$-greedy as done in [2], but in the multi-agent setting, it is unclear whether an attack on CO-UCB provides insight into attacks on other algorithms. Perhaps the authors can further expand on this point - is CO-UCB as fundamental for heterogeneous multi-agent bandits as UCB is for single-agent bandits?
- Figure 3a is not clear. What is the 'average regret' that is plotted here? Since it remains constant, I suppose it should be some kind of instantaneous regret, but this is not explained at all and the only regret defined in this paper is the cumulative one. In addition, it is unclear how the OA and LTA algorithms are performed without AAS.

[1] Zuo, Shiliang. "Near Optimal Adversarial Attacks on Stochastic Bandits and Defenses with Smoothed Responses." International Conference on Artificial Intelligence and Statistics. PMLR, 2024.

[2] Jun, Kwang-Sung, et al. "Adversarial attacks on stochastic bandits." Advances in neural information processing systems 31 (2018).

**Questions:**

- In Table 1, do we assume that all arms have different means? Otherwise, it is possible that the same agent has several optimal arms and is counted in $M_{*}(k)$ for several arms. The same holds for the results in Thms 3,4.
- In line 359 for $\hat{\mu}_{t}(k)$, why are all pre-attack values $X_t^{(m',0)}(k)$ for arm $k$ counted? If I understand correctly, it does not hold that all agents draw the same arm at each round.
- For the heterogeneous setting, do the authors think it is possible to choose a smaller group of target agents than the one in this paper? Can the attack cost be lower than $O(K\log T)$ for some other attack strategy?
- In line 203, can the authors expand more on why the multi-agent setting is more challenging, and what makes the analysis more involved?
- For Figure 3a, what is the average regret, and how were experiments for OA and LTA performed without AAS?
- In the discussion on page 8 before 4.2.4, can you explain in more detail what it means that target agents are 'the first' to pull an arm? This point is repeated a few times but is not entirely clear.

---

> ### Author Response · Authors · 2024-12-02
>
> We sincerely thank the reviewer for highlighting the minor issues, which we will thoroughly address in the updated version.
>
> **Q1.** *This work is not written clearly and assumes the reader possesses extensive technical knowledge about adversarial attacks in bandits. For example, the term attack value is not explained in the main text, and the idea behind equation (2) is also unclear. The parameters in Thm 1 are undefined as well: What are $\alpha, \Delta_0$?*
>
> **A1.** Thank you for pointing this out. The term "attack value" refers to the amount added by the attacker to the reward generated from the original distribution. Regarding the parameters in Theorem 1.
>
> - $\alpha$ is a parameter of the UCB algorithm, which controls the width of the confidence interval. Specifically, in the CO-UCB algorithm, the confidence interval for arm $k$ at round $t$ is defined by a radius of $\sqrt{\frac{\alpha \log(t)}{\hat{n}_t(k)}}$, where $\hat{n}_t(k)$ is the number of times arm $k$ has been pulled.
>
> - $\Delta_0$ is a parameter in our attack strategy. It is used to compute the attack value, ensuring that the post-attack empirical mean reward of a targeted arm is manipulated to be less than the empirical mean reward of the desired arm. Intuitively, $\Delta_0$ controls the margin by which the attacker shifts the rewards to influence the agent’s decision-making process.
>
> We will clarify these terms and the underlying ideas behind Equation (2) more explicitly in the revised version to improve accessibility for a broader audience.
>
> **Q2.** *Table 1 is misleading. For the first and second lines, it is unclear whether knowledge about the time horizon $T$ is needed. The authors can cite [1] rather than [2] if it is not. More importantly, the third line assumes oracle knowledge of the attacker about arm rankings (Thm 3), which is a strong assumption. When using the version that does not use this assumption, the attack cost becomes much larger at $O(K\log(T)/\Delta_{\min}^2)$ (Thm 4).*
>
> **A2.** Thank you for your careful review. We will revise Table 1 to explicitly clarify whether knowledge of the time horizon $T$ is required in each case. For the heterogeneous setting, we acknowledge that the attack cost of the LTA algorithm is $O(K\log(T)/\Delta_{\min}^2)$, as shown in Theorem 4. This cost arises due to the rank learning phase. We believe it could potentially be optimized with a more refined design for learning under attack. While this remains an open direction, we will update the table to clearly specify the assumptions made in the heterogeneous setting and include the corresponding results of the LTA algorithm to provide a balanced presentation.
>
> **Q3.** *The problem addressed in this paper could be motivated better.*
>
> **A3.** Thank you for highlighting the concern regarding privacy preservation in multi-agent bandit systems. While privacy is indeed a critical consideration in certain applications involving user data, there are many scenarios where privacy constraints are less stringent or irrelevant. For instance, recommendation systems with multiple servers operated by the same organization often involve internal information sharing, where privacy is not a concern. Similarly, in robotic systems, cooperative tasks such as source search [A, B] represent practical scenarios where agents collaborate without privacy constraints. Moreover, the CMA2B framework has been extensively studied in contexts that do not involve privacy-sensitive data, as demonstrated by prior work (e.g., Vial et al., 2021; Yang et al., 2021, 2022; Wang et al., 2022, 2023a,b). We will revise the introduction to include these examples and references, providing a broader and more practical motivation for the problem.
>
> [A] Shuai Li, Ruofan Kong, and Yi Guo. Cooperative distributed source seeking by multiple robots: Algorithms and experiments. IEEE/ASME Transactions on mechatronics, 19(6):1810–1820, 2014.
>
> [B] Long Jin, Shuai Li, Lin Xiao, Rongbo Lu, and Bolin Liao. Cooperative motion generation in a distributed network of redundant robot manipulators with noises. IEEE Transactions on Systems, Man, and Cybernetics: Systems, 48(10):1715–1724, 2017.
>
> **Q4.** *While the attack design for heterogeneous systems is novel, it should be mentioned that the attack itself used in this paper technically resembles techniques from previous works [1,2] (for instance, equation (2)).*
>
> **A4.**  Thank you for pointing this out. As mentioned in Lines 202 and 223, we acknowledge that the attack value design draws on techniques from [1, 2]. We will explicitly clarify this connection for equation (2) as well. However, the primary focus of this paper is to address the novel challenges posed by cooperative multi-agent learning. We contribute by introducing non-trivial results, including the AAS and TAS algorithms tailored for heterogeneous settings, accompanied by rigorous theoretical analyses.

---

> > ### Author Response · Authors · 2024-12-02
> >
> > **Q5.** *The difficulty of adversarial attacks on a heterogeneous multi-agent system is not fully characterized. Regarding the target agents group, it is unclear to me if the one used in this paper is the minimal one, or if we can use fewer target agents with the same effect. There is no lower bound to show how good this result is.*
> >
> > **A5.** We acknowledge that the target agent group chosen by the TAS algorithm might not always be the minimal group for all problem instances. This is because TAS selects one target agent for attacking each local optimal arm in $\mathcal{K}_0$. However, in the worst-case scenario, the dependency on $|\mathcal{K}_0|$ is unavoidable. For instance, consider a scenario where there is no overlap between the arm sets of agents with different local optimal arms. In such a case, multiple agents must be targeted to ensure a successful attack. For example, if agent group 1 has arm sets {1,2}, {1,3} and agent group 2 has arm sets {4,5}, {4,6}, at least one agent from each group must be selected to achieve the desired attack outcome.
> >
> > Regarding the attack cost in Theorem 4, as noted above, it arises primarily from the rank learning phase, which could potentially be optimized with a more refined design for learning under attack. As for the lower bound, we note that even in the single-agent case, the tightness of the attack cost's dependence on $K$ remains an open problem. However, we agree that the $\log T$ term in our results is worse than the lower bound of $\Omega (\sqrt{\log T})$ for attacking single-agent UCB, underscoring a promising direction for further improvement.
> >
> > **Q6.** *In the heterogeneous case, an attack is only designed for one algorithm. Is CO-UCB as fundamental for heterogeneous multi-agent bandits as UCB is for single-agent bandits?*
> >
> > **A6.** Thank you for raising this point. First, we want to emphasize that the proposed AAS and TAS algorithms are designed to identify agents vulnerable to attacks and select target agents, and their applicability is independent of the specific multi-agent bandit algorithms used. These algorithms can be applied to any CMA2B setup, including those not based on UCB.
> >
> > Second, we believe that CO-UCB serves as a fundamental algorithm for heterogeneous CMA2B systems, as most existing algorithms are UCB-based (e.g., Yang et al., 2021, 2022; Wang et al., 2023b). Our attack strategy is not limited to CO-UCB. It can be extended to other UCB-based algorithms and the overall analytical framework remains consistent across these extensions, with modifications made to the attack values to adapt to the specifics of the targeted algorithm.
> >
> > **Q7.** *Figure 3a is not clear. What is the "average regret" that is plotted here? In addition, it is unclear how the OA and LTA algorithms are performed without AAS.*
> >
> > **A7.** We apologize for the lack of clarity in the figure. The "average regret" (or cost) plotted in Figure 3a represents the cumulative regret (or cost) up to round $t$ divided by the number of rounds $t$. This metric is used to provide a time-averaged view of the cumulative regret or cost.
> >
> > Regarding the OA and LTA algorithms without AAS, their workflow is as follows: the attacker treats all agents as target agents and directly applies TAS and the corresponding attack algorithms to each agent. We will include these clarifications in the revised version.
> >
> > **Q8.** *In Table 1, do we assume that all arms have different means? Otherwise, it is possible that the same agent has several optimal arms and is counted in $M_{\*}(k)$ for several arms. The same holds for the results in Thms 3,4.*
> >
> > **A8.** Thank you for raising this point. While assuming a unique optimal arm for each agent simplifies the analysis, our algorithms do not rely on this assumption. With appropriate AAS and TAS, the attacks can still be successful even when agents have multiple optimal arms. In such cases, the number of affected agents would be the sum of $M_\*(k)$ across all optimal arms, adjusted by subtracting the number of agents that are double-counted due to having multiple optimal arms.
> >
> > **Q9.** *In line 359 for $\hat{\mu}_t(k)$, why are all pre-attack values $X_t^{(m,0)}(k)$ for arm $k$ counted? If I understand correctly, it does not hold that all agents draw the same arm at each round.*
> >
> > **A9.** Sorry for the confusion. You are correct that the pre-attack values $X_t^{(m,0)}(k)$ should only account for those agents that select arm $k$ in round $t$. We will correct this in the next version.
> >
> > **Q10.** *For the heterogeneous setting, do the authors think it is possible to choose a smaller group of target agents than the one in this paper? Can the attack cost be lower than $O(K\log(T))$ for some other attack strategy?*
> >
> > **A10.** Please see A5.
> >
> > **Q11.** *In line 203, can the authors expand more on why the multi-agent setting is more challenging, and what makes the analysis more involved?*
> >
> > **A11.** Please see A1 to Reviewer bPAF.

---

> > > ### Author Response · Authors · 2024-12-02
> > >
> > > **Q12.** *In the discussion on page 8 before 4.2.4, can you explain in more detail what it means that target agents are 'the first' to pull an arm? This point is repeated a few times but is not entirely clear.*
> > >
> > > **A12.** The term "the first" refers to the fact that, the chosen target agents are the earliest within their respective groups (which share the same optimal arms) to pull the attacked local optimal arms once the UCB of these arms exceeds the UCB of the currently highest arm.  As shown in Figure 2a, the attacks generally reduce the UCBs of the local optimal arms, making them less attractive initially. However, as the confidence radius of their UCBs grows over time, these arms gradually become more appealing. Our selection of target agents ensures that they have the lowest mean reward for their second-best arms within their groups. This design guarantees that these target agents are more likely to switch to the attacked local optimal arms sooner than other agents in their groups. This early switch provides sufficient opportunities for the attacker to manipulate these arms effectively.

---

### Official Review · Reviewer_jeLP · 2024-11-04

**Soundness:** 2
**Presentation:** 3
**Contribution:** 1
**Rating:** 3
**Confidence:** 5

**Summary:**

This work considers the cooperative multi-agent multi-armed bandits (CMA2B) problem to design attack algorithms, which aim to attack on a few agents to influence the decisions of the rest. The authors consider two different settings, i.e., (1) homogeneous settings, where agents operate with the same arm set, and (2) heterogeneous settings, where agents may have distinct arm sets, to design different attack algorithms. Then they provide theoretical analysis on the caused bad regret of CMA2B system under their attack algorithms. Finally, they conduct numerical experiments to verify the efficiency of their proposed algorithms.

**Strengths:**

1.	This paper is well-written and easy to follow, with clear assumptions and a thorough review of existing literature.
2.	The authors address both homogeneous and heterogeneous settings in designing efficient attack algorithms, offering a solid theoretical analysis of the regret lower bounds induced by these attacks.

**Weaknesses:**

While the paper is presented well, the assumptions are highly restrictive, and the proposed attack algorithms apply only to specific CMA2B algorithms (i.e., CO-UCB or UCB-TCOM), which limits the general applicability of this work.

Specific concerns include:
(1)	No Collision: The authors overlook collisions in the multi-agent MAB (MMAB) problem, lacking literature support for this choice. Most studies on MMAB, including works like Boursier & Perchet (2019), incorporate collisions when multiple players select the same arm simultaneously. This assumption is particularly relevant in applications such as cognitive networks.
(2)	Pre-Attack Reward Observability: The paper assumes that the attacker can observe pre-attack rewards for all agents, which seems unrealistic in practical applications. It remains unclear how an attacker would access this information.
(3)	Prior Knowledge of Local Arm Sets: In the heterogeneous setting, the assumption that the attacker knows each agent’s local arm set also seems impractical.
(4)	Algorithm Awareness: The assumption that the attacker knows each agent’s exact algorithm is the strongest and most limiting. This could be plausible if the attacker were an agent within the CMA2B setup, yet targeting a specific algorithm still limits the broader impact of the proposed methods.
(5)	Other strong assumptions include (a) the attacker knows the exact leader in the leader-follower structure, and (b) The attacker has prior knowledge of the reward ranking of all arms. While the authors relax this latter assumption in Section 4.3, introducing this relaxation earlier could strengthen the contribution.
I recognize that this is a theoretical study; however, given that the motivation for these attack algorithms is practical application, the heavy reliance on these assumptions restricts their practical relevance.

**Questions:**

Pls see my questions in weaknesses.

---

> ### Author Response · Authors · 2024-12-02
>
> **Q1.** *While the paper is presented well, the assumptions are highly restrictive, and the proposed attack algorithms apply only to specific CMA2B algorithms (i.e., CO-UCB or UCB-TCOM), which limits the general applicability of this work.*
>
> **A1.** Thank you for highlighting these detailed potential extensions. We acknowledge that many of these points are promising directions for future work. However, before addressing each point individually, we would like to emphasize that the primary objective of this paper is to expose the vulnerabilities inherent in collaborative learning by analyzing representative CMA2B algorithms. Specifically, we have conducted a rigorous study of both homogeneous and heterogeneous cooperative settings, yielding non-trivial theoretical results . We believe our work uncovers core vulnerabilities in multi-agent cooperative bandits, laying a foundation that can inspire future research in this field.
>
> **Q2.** *No Collision: The authors overlook collisions in the multi-agent MAB (MMAB) problem, lacking literature support for this choice.*
>
> **A2.** We agree that addressing the collision problem in MMAB is important, particularly in applications such as cognitive networks. However, it is also crucial to study multi-agent cooperative learning settings without collisions, as highlighted in the CMA2B literature (e.g., Vial et al., 2021; Yang et al., 2021, 2022; Wang et al., 2022, 2023a,b). In this paper, we focus on vulnerabilities introduced by cooperation under these non-collision settings, which align with the assumptions in much of the CMA2B literature. While we acknowledge that incorporating collisions could introduce additional challenges for attack design, we believe the insights presented in our work provide a valuable foundation that can inspire advancements in addressing vulnerabilities in collision-based settings.
>
> **Q3.** *Pre-Attack Reward Observability: The paper assumes that the attacker can observe pre-attack rewards for all agents, which seems unrealistic in practical applications. It remains unclear how an attacker would access this information.*
>
> **A3.** Building on adversarial attacks in single-agent bandits, we adopt the same threat model commonly used in the single-agent literature (e.g., Jun et al., 2018; Liu \& Shroff, 2019; Zuo, 2024). This model assumes that the attacker has access to all pre-attack reward information. However, it is important to highlight that our attack algorithms rely on reward information from the target agents (including shared samples from other agents). Notably, our algorithms can still function effectively even when the set of attackable agents is restricted (footnote 1 on page 3).
>
> **Q4.** *Prior Knowledge of Local Arm Sets: In the heterogeneous setting, the assumption that the attacker knows each agent’s local arm set also seems impractical.*
>
> **A4.** Since the CMA2B model enables agents to communicate with one another, local arm set information can be easily shared across the system, incurring only a constant communication cost. This assumption aligns with the practices in the heterogeneous CMA2B literature (Baek \& Farias, 2021; Yang et al., 2022; Wang et al., 2023b).
>
> **Q5.** *Algorithm Awareness: The assumption that the attacker knows each agent’s exact algorithm is the strongest and most limiting.*
>
> **A5.** In this paper, we primarily focus on representative CMA2B algorithms, assuming that the attacker has knowledge of the agents' algorithms, which enables more efficient attacks. However, as discussed in Section 4.2.4, we also extend our findings to general homogeneous CMA2B algorithms without requiring knowledge of the exact algorithms and provide a cost analysis in Appendix B.5. For heterogeneous settings, AAS and TAS remain effective tools for identifying agents vulnerable to attacks and selecting target agents, even without precise algorithm awareness. Developing more general attack strategies for heterogeneous CMA2B algorithms remains an open and promising direction for future research.

---

> > ### Author Response · Authors · 2024-12-02
> >
> > **Q6.** *Other strong assumptions include (a) the attacker knows the exact leader in the leader-follower structure, and (b) The attacker has prior knowledge of the reward ranking of all arms.*
> >
> > **A6.** (a) For the leader-follower algorithm, the actions of the two types of agents are sufficiently distinct, allowing the attacker to quickly identify the leader through observations at the beginning of the game, even without prior knowledge. While we did not explicitly include this identification phase in our algorithm, our primary goal was to highlight the vulnerability of the leader within the framework.
> >
> > (b) Regarding the reward ranking assumption, our oracle attack in Section 4.2 relies on this knowledge for theoretical analysis, while it is relaxed in Section 4.3 with the LTA algorithm. We introduce these two algorithms sequentially to first provide key theoretical insights from the oracle attack and then demonstrate the more general applicability of the LTA algorithm. We appreciate the reviewer’s suggestion and will revise the manuscript to introduce and emphasize the LTA algorithm earlier to strengthen the practical relevance of our work.

---

### Official Review · Reviewer_bPAF · 2024-11-04

**Soundness:** 3
**Presentation:** 3
**Contribution:** 2
**Rating:** 6
**Confidence:** 4

**Summary:**

This paper studies adversarial attack strategies in the Cooperative Multi-Arm Bandit setting CMA2B. In the Cooperative Multi-Arm bandit setting ($M$) agents pull one arm each in round $T$ and observes reward sampled from a distribution associated with the arm. The agents some can communicate with each other to accelerate learning. In this paper, the authors study how an attacker can take advantage of agents trusting other agent's data, and can attack a subset of agents and influence other agents as well.
They study the problem in two settings:

i) Homogeneous setting: Here all agents have access to all arms. They show only one agents can be injected with $o(T)$ corruptions and cause a target suboptimal arm to be selected for $T - o(T)$ rounds under the CO-UCB algorithm. They also show results for UCB-TCOM (a communication efficient UCB style algorithm), DPE2 (a leader-follower algorithm), and general no-regret algorithms.

ii) Heterogeneous setting: In this setting, each agent has access to a subset of arms. They show that there can be instances where arms are distributed in a way that to incur linear regret to each agent in CO-UCB, an attacker would need a linear attack budget.

They propose a selection algorithm called AAS to select the arms that are optimal for as many non-conflicting agents and TAS to select agents to attack. They provide regret guarantees for the CO-UCB algorithm under this setting.

They also provide experiments with simulated data for comparing their attacks on the CO-UCB algorithm.

**Strengths:**

1. The setting of adversarial attacks on cooperating bandits studied in the paper is well-motivated
2. The authors provide the analysis for the homogeneous and heterogeneous settings, and under different CMA2B algorithms, at least in the homogeneous setting. The results that UCB style CMA2B algorithms can be  $o(T)$ corruptions and cause a target suboptimal arm to be selected for $T - o(T) times by targeting just one arm in homogeneous settings has novelty. The detailed analysis of the heuristics AAS and TAS also shows that agents showing that a large number of agents can be attacked using sublinear corruptions is also valuable. The authors also provide an analysis of the case when rankings are not known to the attacker making the attack more practical.
2. The authors provide experimental evidence for the novelty of AAS and TAS.
3. The paper is overall well written, with clear notations

**Weaknesses:**

1. The homogeneous setting studied under the Co-UCB is very similar to the results for UCB obtained in Jun et al. [2018]  as every agent can see every other agents rewards and updates the mean rewards and counts, it's similar to the one agent setting, but the agent obtains $M$ samples in every round from the arm they pull instead of 1, thus I believe the results of \citet{jun2018adversarial} can can be modified to obtain this result, so there is limited novelty in this result.

2. Most of the results are on UCB style algorithms, or on the communication model where all agents immediately share the information will all other agents. I feel this is fairly limited setting as the samples are effectively shared easily and it makes the attack easy as any agent can be attacked with a big enough value that gets propagated to all other agents. What happens when the communication model is constrained through addition structures for e.g., the graph structured discussed in Landgren et al. [2016].
They do provide analysis for UCB-TCOM and DPE2 with delayed communication, but in both case, the information is still easily transmitted to all agents.

3. I believe the results against general algorithms need some more refinement in terms of assumptions. Line 1308 on Page 25 assumes that since the other arms' samples always come from a distribution with a mean less than the mean of the target arm, we can consider arm $K$ to be the best arm. The results for general bandit algorithms are based on the fact that reward samples are being sampled from a fixed distribution, if the sample distribution for the other arms is changing with $t$ (as the attack), I don't think we can directly claim Assumption 2 to hold. The main point is that in Assumption 1, it's not clear what it means that the 'agents maintain identical empirical means for making decisions'. Maybe the authors can refine Assumption 1 to a class of more restricted MAP/CMA2B algorithms like the mean based algorithms defined in Xu et al. [2021].

4. It's difficult to understand the $T_0$ term in the heterogeneous results, and how it's always a constant.

5. The experiments provided are fairly limited, with no error bars.

Minor comments:
In line 707, it should be $- \hat{\mu} (K)$ instead of  $+ \hat{\mu} (K)$

References:
1. Kwang-Sung Jun, Lihong Li, Yuzhe Ma, and Jerry Zhu. Adversarial attacks on stochastic bandits. Advances
in neural information processing systems, 31, 2018.

2. Peter Landgren, Vaibhav Srivastava, and Naomi Ehrich Leonard. Distributed cooperative decision-making
in multiarmed bandits: Frequentist and bayesian algorithms. In 2016 IEEE 55th Conference on Decision
and Control (CDC), pages 167–172. IEEE, 2016.

3. Yinglun Xu, Bhuvesh Kumar, and Jacob D Abernethy. Observation-free attacks on stochastic bandits.
Advances in Neural Information Processing Systems, 34:22550–22561, 2021

**Questions:**

1. What are the additional challenges in adapting the results of Jun et al. [2018] for the homogeneous setting with CO-UCB?

2. Can we experimentally verify if these results would work for cooperative versions of simple MAB algorithms like $\epsilon$-greedy or Thompson sampling?

---

> ### Author Response · Authors · 2024-12-02
>
> **Q1.** *Limited novelty of the results in homogeneous settings.*
>
> **A1.** We acknowledge that the analysis of attacking CO-UCB in the homogeneous setting shares similarities with the single-agent UCB analysis in Jun et al. [2018] and Zuo [2024]. However, we would like to highlight the following points: (a) We are the first to formally prove the cost of attacking homogeneous CO-UCB and clearly position this as a warm-up result, as presented in Table 1. Our result demonstrates that it is sufficient to ruin the entire multi-agent system by attacking only one agent in homogeneous settings. This finding provides a baseline for comparison with the heterogeneous setting and highlights the vulnerabilities inherent in cooperative multi-agent systems. (b) We prove that our attack policy remains effective against more sophisticated homogeneous algorithms, such as those with delayed feedback or leader-follower structures. These results, detailed in Appendices B.3 and B.4, address new challenges specific to these scenarios, such as managing asynchronous reward updates and accounting for the distinct roles of leaders and followers. We hope these points clarify the novelty and relevance of our results in the homogeneous setting.
>
> **Q2.** *What happens when the communication model is constrained through addition structures?*
>
> **A2.** We agree that exploring communication models with graph structures, such as those discussed in Landgren et al. [2016], is an important and promising direction for future research. In such scenarios, identifying the most impactful target agents for attack becomes more complex, as the influence of each agent on the cooperative learning system would depend on the graph topology and connectivity. While our current work does not address these more complex communication graphs, our analysis in the heterogeneous settings provides insights into identifying target agents based on their relative importance and influence within the cooperative framework.
>
> Additionally, we would like to emphasize that the communication model impacts both the effectiveness of cooperative learning and the vulnerability to attacks. When communication is constrained (e.g., by a graph structure), it is true that attacks may become more challenging. However, the benefits of cooperation would also diminish due to limited information sharing. This highlights an intriguing trade-off between cooperation efficiency and system robustness. The primary goal of this paper is to uncover vulnerabilities in fully collaborative systems, and we hope our findings can motivate further research into addressing these trade-offs in more constrained communication settings.
>
> **Q3.** *Refinement of assumptions for attacks against general algorithms.*
>
> **A3.** We apologize for any confusion caused by Assumption 1 and sincerely appreciate the reviewer’s suggestion to reference Xu et al. [2021]. We agree that refining Assumption 1 to explicitly consider a class of mean-based CMA2B algorithms, such as those defined in Xu et al. [2021], would provide better clarity and a more precise scope for our analysis. Regarding Assumption 2, we acknowledge the concern about the changing sample distributions under attack. The post-attack distribution can be viewed as a shifted version of the pre-attack distribution with a gap of $\Delta(k, K) + 4\beta(1)$. In this context, we expect general bandit algorithms would still identify $K$ as the best arm and pull suboptimal arms limited times. We will explicitly clarify these assumptions in the updated version.
>
> **Q4.** *It's difficult to understand the $T_0$ term in the heterogeneous results, and how it's always a constant.*
>
> **A4.** $T_0$ represents a threshold that ensures our selected target agents consistently pull the arms intended for attack before other agents, thereby providing sufficient opportunities for successful attacks. Without this threshold, other agents might collect too many unattacked samples, which would increase the attack costs of the target agents.
>
> $T_0$ is a constant since it is determined as the feasible solution of Eq. (3) with respect to $t$, where the right-hand side is a problem-dependent coefficient that does not vary with $t$. For a more detailed explanation, please refer to Lines 383–391.
>
> **Q5.** *Experiments are limited with no error bars. Can we experimentally verify if these results would work for cooperative versions of simple MAB algorithms like $\epsilon$-greedy or Thompson sampling?*
>
> **A5.**  Thank you for your valuable suggestions regarding the experiments. We will include error bars in the updated version and extend our experiments to validate the effectiveness of our attack strategies on other simple cooperative MAB algorithms.

---

### Meta-Review · Area_Chair_MJjN · 2024-12-21

**Metareview:**

This paper studies adversarial attacks on cooperative multi-agent multi-armed bandits (CMA2B) in both homogeneous and heterogeneous settings. The reviewers agree that the paper addresses an interesting and well-motivated problem. However, the reviewers raised several shared concerns including limited technical novelty, unclear discussion about contributions, relatively strong assumptions, and presentation issues with technical results. I recommend rejection for current version, and believe the submission would get much stronger by revising according to reviewers' suggestions.

**Additional Comments On Reviewer Discussion:**

The reviewers raised several concerns about limited technical novelty, unclear discussion about contributions, relatively strong assumptions, and presentation issues with technical results. The authors responded by clarifying the difference and novelty with related works (some of which were ignored), explaining the contribution especially the heterogeneous setting, clarifying assumptions, and identifying some typos/errors in technical results. Overall, the paper has the potential but would require substantial revision. Thus, my recommendation is rejection.

---

### Decision · Program_Chairs · 2025-01-22

Reject